# Cause of Death in Charred Bodies: Reflections and Operational Insights Based on a Large Cases Study

**DOI:** 10.3390/diagnostics12081986

**Published:** 2022-08-16

**Authors:** Aniello Maiese, Costantino Ciallella, Massimiliano dell’Aquila, Alessandra De Matteis, Chiara Toni, Andrea Scatena, Raffaele La Russa, Eleonora Mezzetti, Marco Di Paolo, Emanuela Turillazzi, Paola Frati, Vittorio Fineschi

**Affiliations:** 1Department of Surgical Pathology, Medical, Molecular and Critical Area, Institute of Legal Medicine, University of Pisa, 56126 Pisa, Italy; 2Department of Anatomical, Histological, Forensic and Orthopedical Sciences, Sapienza University of Rome, Viale Regina Elena 336, 00161 Rome, Italy; 3Section of Legal Medicine, Department of Clinical and Experimental Medicine, University of Foggia, 71122 Foggia, Italy

**Keywords:** charred bodies, flame injuries, forensic protocol, forensic medicine, smoke inhalation, carbonized bodies

## Abstract

Our study aims to demonstrate the experience of analyzing fully or partially charred corpses to offer a proper implementation protocol for determining the cause of death. In this study, we present a total of 103 cases obtained from the University of Rome La Sapienza and the University of Pisa archives. All cases were classified based on the extent and severity of burns using a visual method. We divided all cases into two groups. The first group included grade I–II burns (21 cases) without the need for identification. The second group (82 cases) included injuries worse than grade burns II, so all cases were analyzed using an analytical method. For each case, we have documented which of the following analyses have been used and the corresponding findings: inspection, autopsy examination, imaging examination, genetic and toxicological examinations, and histological examination. The results describe the main diagnostic findings and show that only the application of all the above systematic analyses can provide greater accuracy and reliability in describing the causes of death or solving problems, such as identification. In conclusion, we propose an available protocol that defines the main steps of a complete diagnostic pathway that pathologists should follow daily in studying charred bodies.

## 1. Introduction

The retrieval of a body drawn by flame exposure often presents a real challenge to the forensic pathologist, as it presents him or her with several questions complicated by the marked alteration/destruction of tissues or, in some cases, the complete mutilation of certain segments of the soma so that even personal identification becomes problematic. Indeed, the action of the flame can significantly alter the characteristics of certain thanatological findings, making diagnostic interpretation very difficult, and new findings can be very similar in their characteristics to injuries of other types (skull fractures, presence of extradural blood leakage, stab wounds, etc.).

Heat and flame, when they act upon the human body, may produce local and general reactions, the severity of which varies in direct proportion to the extent and depth of the burn produced, and to the temperature to which the body is exposed.

The minimum temperature necessary to cause damage to the human organism is represented by an effective skin temperature of 44 °C, in which conditions of not less than 6 h are necessary to achieve a second-degree burn [1,2,3]. Between 44 °C and 51 °C, a 1 °C increase in temperature halves the exposure time required for skin damage. Above 51 °C, excess heat is no longer dissipated by convection through the skin capillaries. However, it should be assumed that any temperature will cause a certain progression of carcass burns. Therefore, when working backwards from an analysis of the cadaveric lesions, it is possible to estimate approximately the duration of the fire that produced them.

The causes of death due to exposure to flames can generally be divided into death directly at the scene of the fire and death after exposure to the flames [4,5,6]; the underlying causes of these types of death are shown in the following figure (Figure 1).

In cases of death by exposure to flames, there is usually evidence of the viability of the heat-related injuries. However, there are also cases where death has been caused by heat and flames without any evidence of viability being found at the inquest. In the literature, the incidence of this type of death is reported to be 3 to 10% of all fire-related deaths. The following causes should be considered:Cyanide toxicity—depending on the material burnt, cyanide toxicity may develop very rapidly, so elevated COHb (carboxyhemoglobin) concentrations may not be detected;Deflagration—death may occur by respiratory arrest due to laryngeal spasm, bronchial spasm, vagus reflexes, or heat shock on inhalation;Oxygen deprivation—deprivation of oxygen at the source of the fire can lead to death;Heat shock—a redistribution of circulating blood volume due to the heat exposure of the skin;Heat rigor—impairment of respiratory function due to sudden thermal rigidity in the chest.

According to Richards [4], at a temperature of 680 °C, the arms show severe signs of charring after 10 min, the legs after 14 min, the bones of the face and arms after 15 min, the tibia and fibula after 25 min, the ribs and skull after 20 min, and the thighs after 35 min. Extensive observations of skull and tooth destruction were made during the cremations. Further observations on the time course of the destruction of whole human bodies during cremations at temperatures between 670 °C and 810 °C were published by Bohnert et al. (1998) [5]. The bodies exhibited a combative posture after about 10 min; after 20 min, the cranial vault was devoid of any soft tissue and cranial heat fractures were evident. At about 30 min, the body cavities became visible, resulting in the exposure of the organs therein, and at 40 min after the commencement of cremation the internal organs were much shrunken and showed a spongy reticular structure. After about 50 min, the extremities were largely destroyed, leaving only the torso, which was also burned after 1.5 h. The complete combustion of a human body required about 2–3 h.

We present a total of 103 cases obtained from the case archives of the University of Rome La Sapienza and the University of Pisa. All cases have been closed by the authorities and have already been filed. Each case was investigated by one or more of the following methods: inspection, autopsy, imaging studies, genetic and toxicological studies, and histological studies.

Our goal was to review the results obtained to create a protocol that highlights the basic steps that the forensic pathologist must take to determine the cause of death in a charred victim.

## 2. Materials and Methods

To establish an operational and methodological protocol to identify a procedure to be followed in case of detection of deaths by fire, the databases of several technical advisors of the Public Prosecutor’s Office of Rome and Pisa (belonging to the Legal Medicine Institute of Rome, La Sapienza and of the University of Pisa) were analyzed. In the research, the words “flame”, “burns”, “burnt”, and “charred” were used so that a total of 103 cases could be selected.

All cases were divided into two groups according to a visual method based on the location, area, depth, and distribution of the burns (first to fourth-degree burns and complete charring) and the need to establish the vitality of the lesions or identify the body. The first group [21 cases] included first-degree burns and second-degree burns that did not require identification of the body or analysis of the vital lesions. These cases were excluded from the study. The second group [82 cases] included cases with complete charring and second-, third-, and fourth-degree burns or burns that needed to be identified (Figure 2).

To establish a standardized protocol, we examined which of the following examinations were performed in each case and what findings were obtained:Inspection;Radiological examination;Autopsy examination or external examination;Histological or immunohistochemical examinations;Toxicological examinations;Genetic examination.

It is necessary to explain that the uniqueness of each case and the Italian legal and procedural structure make it impossible to guarantee the application of all the above procedures for each case.

### 2.1. Inspection

During the forensic inspection, the condition of the environment was described, including the type, quantity, and distribution of ignited or highly heated materials, such as red-hot metal material and molten glass or aluminum, or traces of building materials, were observed. An analysis of the clothing was also carried out, showing the location, extent and direction of the damage caused by the heat exposure, but also the size, color, and types of clothing, such as items in pockets and watches if the charred body was not identifiable. Possible personal items near the scene were recorded.

### 2.2. Radiological Investigations

When possible, a full-body postmortem computed tomography (PMCT) scan was performed within 24 h of the forensic examination or admission to the morgue.

Generally, a preliminary postmortem whole-body examination was performed using a 16-detector multislice CT prior to a conventional autopsy. The bodies were placed supine on the CT table. The average CT scan time was 11.5 min (range 8–15 min). The CT scans obtained were analyzed using the open-source software OsiriX on a Mac OS X computer, which provides a 3D representation of the DICOM images.

In all cases where PMCT was not available, conventional whole-body radiographs were obtained.

### 2.3. Autopsy Examination or External Examination

Autopsy examinations and external examinations were performed at the department which specialized in necropsy in the Sapienza University in Rome.

In both procedures, the external examination of the bodies was based on observation of the localization, area, depth, and distribution of the burns. In each case, the state of piliferous formation (hair, eyebrows, and beard), the change in color of the skin (“leopard skin”), the presence of blisters, and areas of erythema, smoke, or charring were described. All skin lesions were carefully photographed, measured, and catalogued. On the head, the accentuation of “crow’s feet” in the periorbital area and sooty mold in the nostrils and oral cavity were examined. On the hand, the presence of adherent materials of various types, such as nail bed debris, was examined. An analysis of the hypostasis (if present and palpable) was performed.

In all cases where identification was uncertain, individual characteristics, such as height, weight, hair and eye color, surgical scars, sexual characteristics, amputations, piercings, or tattoos were documented. Dental elements were removed, photographed, and numbered, as well as in the case of dentures or possible personal items found on the body.

In the autoptic procedure, the standard protocol was applied, with the particularity that the airways were removed “en bloc”. After isolation of the esophagus and aorta, the tongue, soft palate, pharynx, hyoid bone, larynx, trachea, bronchi, and lungs were carefully dissected “en bloc”.

### 2.4. Histological and Immunohistochemical Examination

Damaged tissue samples were collected and fixed in formalin, secondarily embedded in paraffin, and finally cut into 3–4 μm thick sections. Hematoxylin and eosin staining (HE) were performed according to the Dako protocol [6]. After preparation of sections of paraffin-embedded tissue, immunohistochemical (IHC) analysis was performed, including fibronectin, CD62P (P-selectin 26P), HSP27 (heat shock proteins 27), HSP90 (heat shock proteins 90), HSP70 (heat shock proteins 70), tryptase, and HIF-1α (hypoxia-inducible factor 1-alpha) but also antibodies, such as anti-HSP27 (heat shock proteins 27), anti- HSP70 (heat shock proteins 70), anti-HSP90 (heat shock proteins 90), and anti-tryptase.

Samples were fixed overnight in phosphate-buffered 4% paraformaldehyde (pH 7.4) at 4 °C. After extensive washing in 0.1 M phosphate-buffered saline (PBS; pH 7.4), the samples were embedded in paraffin and cut into transverse skin sections using a microtome (HistoCore AUTOCUT R, Leica microtome).

### 2.5. Toxicological Examination

All toxicological analyses were performed at the Institute of Forensic Medicine, Department of Toxicology, Pisa.

Carboxyhemoglobin concentration (COHb) and hydrocyanic acid (HCN) concentration were calculated in the central blood using a spectrophotometric method. Screening for substance abuse and alcohol was performed in available biological fluids (central and peripheral blood, urine, and/or stomach contents) or tissues (hair, pubic hair, brain, liver, and kidneys) using liquid chromatography–tandem mass spectrometry (LC–MS/MS).

### 2.6. Genetic Investigation

For the genetic study, samples were taken from uninjured areas and analyzed in the genetic laboratory of the University of Pisa. When available, fungal formations, saliva, and fragments of muscle tissue or teeth were collected during autoptic examination and used for DNA analysis.

Here, DNA was extracted using the EZ1 Advanced and the EZ1 Tissue Kit (QIAGEN, Hilden, Germany) and quantified using a spectrophotometer at 260 nm. A PowerPlex^®^ 18D was used as a multiple STR amplification kit,. Electrophoresis was performed on a 3500 Genetic Analyser (Life Technologies, Carlsbad, CA, USA) using an interpretation threshold of 110 RFU after validation of the internal kits.

### 2.7. Diagnosis of Death

In the end, the different causes of death were compared and divided into the following three groups: unidentified, indirect diagnosis, and direct diagnosis. The subdivision criterion was based on the presence or absence of a diagnosis of exclusion, so that all diagnoses that did not directly describe the cause of death required the involvement of multiple organs and apparatuses to be explained.

Thus, causes of death, such as “cardiorespiratory failure”, “multi-organ failure”, and “respiratory failure”, were described, which are generally “indirect” diagnoses and, therefore, considered less specific. Conversely, all diagnoses that contained a definite diagnosis were considered “direct diagnoses”.

Subsequently, the data were divided into those in which a radiological and toxicological examination had been performed (R + T) and those in which none or only one of the two examinations had been performed (No. R + T).

## 3. Results

Of the total 82 cases, 53 were male and 29 were female; the average age was 43 years. Regarding the degree of burns of the bodies affected by flame exposure, 36 cases showed extensive burns (of different degrees), while 46 cases showed extensive charring to complete charring. Table 1 provides a graphical representation of the assay performed in each case. In 46 cases, the corpses were completely charred, while in the remaining cases, different degrees of burns were documented; in particular, in 16/82 cases, burns of degrees II to IV were described, as shown in Figure 3.

As shown graphically in Figure 4, histological analyses were performed in 44 cases, similar to the toxicological examinations (45/82). Radiological and genetic examinations were rarely performed, and barely reached a quarter of the cases. All examinations were performed in only in 4/82 cases.

A forensic examination was performed exclusively in 22 cases; of these, 8 had burns and 14 were severely charred, as shown in Table 2. Cases 18 and 19 described firearm injuries, while seven cases were aviation accidents. In case 44, a charred male body was found in a bed with the left wrist raised and tied to the headboard. A coin covered the left eye and another coin lay next to the right eye. Figure 5 shows some scenarios of the forensic examination.

Radiological investigations were carried out in 18 cases (15 CT and 3 Rx); of these, 3 were carried out on burnt bodies (1 Rx and 2 CT), and 15 on charred bodies (2 Rx, 13 CT), as described in Figure 6. Bone fractures were found in eight cases and metallic material was found in five cases. In one case, the metallic body found was clearly identified as an earring. Cerebral lesions (subdural hematoma, epidural hematoma) were found in only two cases. A body was dismembered. Figure 7 shows some CT total body reconstructions on partially charred cadavers. The CT recognised skull fractures, fractures of the mandible, ribs, sternal fractures, L4 and L5 body fractures, arm fractures, and heat amputation of the legs. Table 3 lists the instrumental findings and the type of radiological examination.

Autopsy examination was performed in 66 cases, 22 of which were burned bodies, while 43 were charred bodies. In only 16 cases was only an external examination performed (14 cases of burned corpses and only 2 cases of charred corpses). Figure 8 shows some of the autoptic findings, such as a tongue covered with blackish material protruding from the oral cavity and trapped between the dental arches, or a thin extradural hematoma. The trachea and bronchi exhibited edematous and hyperemic cherry red mucosa with sooty residue inside and sooty fragments.

As shown in Table 4, autoptic examination allowed the identification of vital lesions according to macroscopic criteria. Postmortem flame exposure was identified in 14 cases of charred cadavers, whereas in 12/26 cases the victim was alive when exposed to flame.

Histologic examination (Table 5) was performed in 44 cases and immunohistochemical examination was carried out in 16 cases. Of the histological examinations, 8 were performed on completely burned cadavers, while immunohistochemical examinations were not performed in any of these cases. In the remaining 36 cases, histological examinations were supplemented by immunohistochemical examinations in only 16 cases. Most immunohistochemical studies were performed on lung tissue, and only case 17 used skin on the neck. In addition, case 17 is the only one in which the expression of HIF-1α, HSP27, HSP90, and tryptase was examined. The HIF-1α showed positive in lung tissue, whereas HSP90 and tryptase were negative only in neck skin tissue. In addition, anti-HSP90 and anti-tryptase showed only weak positivity in facial skin.

Figure 9 shows some histological findings. The trachea preparation shows endoluminal incongruent blackish material that is probably sooty and adherent to the epithelium. The proximal portion of the larynx shows heat-induced epithelial changes that are readily seen by the antibody reaction to HSP 27. Immediate death reveals debrided and coagulative necrosis of the membranous mucosa in the nasopharynx, larynx, and trachea. Blackened particles were found in the bronchial structures and terminal bronchioles but, also more sporadically, in the alveoli. Obstruction, oedema, and alveolar hemorrhage were the most common histologic findings. Fibronectin was examined in 14 cases, but it was found to be negative in immunohistochemistry in only 2 cases. Notably, in case 1, Hps70 was weakly positive in the presence of negative CD62P. In contrast, in case 2, when fibronectin was negative, the behavior of the other two markers seemed to be vehemently opposite. In the remaining 12 cases, the positivity of fibronectin always corresponded to the positivity of Hps70 and CD62P.

Toxicological studies (Table 6) were performed in 45 cases. Of these, 11 were performed on burned corpses, and 34 were performed on charred corpses.

Analyzing only the CoHb concentration, as shown in Figure 10, 35/45 cases had CoHb concentrations between 10 and 50%, while only 8 cases had CoHb concentrations above 50%. Toxicological screening for drug or alcohol abuse was performed in only one case, but CoHb or HCN were not examined.

Only nine cases resulted negative for COHb (carboxyhemoglobin) or HCN (hydrogen cyanide) poisoning.

Toxicological screening showed that most cases tested were negative for substance abuse or alcohol. The alcohol concentration in all cases was studied in blood, with 4/45 cases testing positive. Figure 11 summarizes the results of the toxicological analysis.

In case 3, cocaine was detected in urine and blood, and benzoylecgonine was also detectable (250 ng/mL in blood). Venlafaxine and O-desmethyl-venlafaxine were positive only in case 38.

Genetic identification studies were conducted in 20 cases, 16 of which were largely charred corpses. In four cases, II–III–IV-degree burns were present. In all cases, it was not possible to determine the identity of the corpse by other means.

Regarding the causes of death in cases of life-threatening injuries due to exposure to flames, in most cases, death was due to carbon monoxide poisoning (20 cases). In the remaining cases, death occurred due to multi-organ failure or septic shock resulting from burn disease (21 cases) or, in a smaller number of cases, due to acute cardiorespiratory failure resulting from flame exposure (6). As shown in the table above, the cause of death could not be determined in two of the cases we analyzed (cases 56 and 82). Case 56 involved body remains found under the debris after a railroad accident, which can be defined as a mass disaster. Case 82 involved completely charred remains found after a traffic accident. In both cases, only the external examination was performed. In five cases, death was due to a polytraumatic event, whereas in cases 1, 4, and 26, death was due to violent mechanical asphyxia resulting from strangulation. In three cases (18, 19, 44), osteo-visceral injuries from gunshots to the neck, thorax, and pelvic region resulted in death. In most cases, severe pathophysiologic sequelae of deep burns affecting the body surface were responsible for death, with varying forms of severity. In four cases, the autopsy revealed acute myocardial infarction, whereas in cases 56 and 82, the cause of death could not be determined.

A full investigation was performed in only four cases, whereas carbon monoxide intoxication was detected exclusively in those cases in which toxicological analyzes were performed. According to an external investigation (14), the cause of death could be related exclusively to multi-organ failure in patients with II, III, and IV-grade burns on almost all body surfaces due to flame exposure. In only two cases did cranioencephalic lesions described by the radiologic examinations result in death. Table 6 provides an overview of all causes of death and the type of analyzes performed. We analyzed the vitality of the lesions only in the charred cadavers. Of the 46 charred corpses, 24 had vital lesions caused by the heat, and 20 were dead the moment they were hit by the flame (lack of vital lesions). In two cases, it was not possible to determine whether the lesions were vital or not. In some cases of antemortem injury (11), vitality could be determined only by macroscopic observation. In all other cases, histologic examination had to be performed.

In addition, Table 7 lists the bodies affected by burns, indicating the degree of burns and the type of death that occurred (immediate or delayed). Determination of the time of death was performed in 36 cases, all of which were burned but not completely charred bodies. In 10 cases, death occurred immediately, regardless of the severity of the injuries. As shown in Figure 12, subjects died in only four cases, but only in one case due to burns. The remaining 24 died later, mainly due to multi-organ failure (18 cases), and only in 1 case due to a septic process.

Both radiological and toxicological analyses were employed in 13/82 (15.2%) cases. However, in the remaining 69, only one or neither examination was performed.

Of the subjects who underwent both radiological and toxicological examinations, direct diagnoses were made in 61.5% of cases, compared with 46.4% of cases in which direct diagnoses were not made. On the other hand, in the cases in which the examinations were not performed or were performed only partially, the indirect diagnoses correspond to 36/82 cases, or 52.2%. On the other hand, in the cases in which they were performed, the percentage is 30.7% (Table 8 and Figure 13).

## 4. Discussion

First, it is necessary to correlate the data obtained by inspection of the cadaver with the data obtained by analysis of the examination site. Analysis of the hypostasis (if present and palpable) can generally provide an initial clue, since the presence of a bright red color (tending toward cherry red) indicates the presence of high levels of carbon monoxide and carboxyhemoglobin in the blood.

In our investigations, we have found that cadavers exposed to flame often have lesions that cannot be clearly interpreted. These are often continuous lesions (without vital signs) that vaguely resemble stab wounds because of the retraction of the skin tissue (“splitting”). These lesions should be carefully photographed, measured, and catalogued. Heat fractures of the long bones are also common, caused partly by muscle retraction and partly by a reduction in bone consistency due to burns (calcination), but also as a result of handling maneuvers.

On the other hand, in the cases of cadavers that we analyzed that were attracted by flame exposure, PMCT allows documentation of injuries typically associated with heat exposure, such as epidural hematomas, fractures, and hepatic and cardiac gas emboli. If the cadaver is severely charred, so-called “heat fractures” are often found, distinguished from those of a traumatic nature by the presence of exposure, a transverse process, and regular margins. Additionally, CT images also provide a useful guide to the location of liquid blood draws to perform blood sampling, which is critical for assessing COHb levels [7,8,9].

The face and skull are among the earliest and most affected areas, and the typical heat fractures that develop in the cranial vault have typical features of PMCT, namely that they are thin, linear, and superficial. Another finding for which PMCT can provide useful information before autopsy is the so-called “epidural pseudohematoma” or “epidural heat hematoma,” which results from retraction of the dura and leakage of blood from the venous sinuses into the epidural space. The epidural heat hematoma has similar characteristics to subdural hematoma (a low density, “crescent-shaped arrangement”), whereas hematoma of traumatic origin has high density and a “lenticular arrangement” Additionally, PMCT is of great importance in the evaluation of gunshot wounds, in the identification of traps, and in the evaluation of the entry and exit wound at the site of the bullet tip (highly radiopaque) [10,11,12].

An advantage of these methods is the absence of secondary damage, which is often a problem with charred corpses [13,14,15,16,17,18,19,20]. It can also provide evidence that the person was breathing during the fire and, therefore, was still alive. The first sign of asphyxia could be a tongue protrusion (best seen in sagittal reconstruction). Bohnert and Hejna, in a retrospective analysis of 61 fire deaths, did not find a statistically significant increased incidence of tongue protrusion in deaths with vital heat exposure [21]. In addition, inhalation of boiling gas leads to mucosal damage, resulting in pulmonary oedema, which has the same characteristics as classic pulmonary oedema (a diffuse “frosted glass” opacity of the lungs). This oedema must also be distinguished from hypostatic pulmonary oedema, in which the “frosted glass” appearance affects the declined areas.

Conventional radiographs are often used to assist in the autopsy, not only to document fractures or typical injury patterns, but also to locate foreign bodies and/or metal fragments; radiographs can be essential in determining ethnicity (by analyzing skull bones, long bones, etc.), sex (by examining skeletal differences, such as the pelvis), age (by examining ossification nuclei), and stature (by analyzing long bones). Furthermore, PMCT can also provide useful information about the presence of personal items and reveal the presence of prosthetic or medical devices in the body (vascular prostheses, osteosynthesis patches/screws, implantable defibrillators, prosthetic joints, pacemakers, etc.). Radiology also plays an important role in forensic identification when examining dental processes. In the analyzed cases, CT and the radiographs showed typical fracture results of heat lesions and fractures due to trauma, which were recorded thanks to the performance of forensic examination and autopsy.

Referring to the protocol of NASA [20], PMCT examination is recommended in all cases of gunshot wounds, charred remains, infants, trauma, and explosions. Our study showed that performing radiological examinations leads to a definite diagnosis in more than half of the cases. For example, in our study, two gunshot wound deaths were identified by PMCT before autopsy was performed.

As for the cases of bodies killed by flame exposure, the autopsy examination of a fire victim is generally aimed at determining the typical signs of flame injury (charring of skin and clothing, possible flame amputations, and a typical “fighter” posture), ruling out other possible causes of death, and looking for signs indicating vitality at the time of the fire. Findings indicative of vitality at the time of flame exposure include inhalation, ingestion of soot, and thermal respiratory injury.

In contrast, findings indicative of postmortem exposure to the flames include hemorrhage, stab wound-like permanent injuries (particularly in the flexor folds of the joints), heat-induced disarticulations, bone fractures, calcifications of the areas affected by the flames, and injuries not caused by the flames. The charred corpse generally exhibits a reduction in the volume of organs and overall body surface area. The cornea has an iridescent appearance and may mimic a blue iris. Where the internal cavities are not directly involved, the viscera are well preserved and have a recognizable structure. The muscles and heart look like “cooked meat”, and the blood is thickened into clots. In this context, great importance should be given to the examination of the respiratory tract, especially regarding the amount of soot. If confined to the pharynx and trachea, it may be associated with postmortem diffusion; the presence of soot particles in the bronchi and especially in their peripheral branches indicates active inhalation of smoke and, consequently, vital injury [21].

A fundamental step in an autopsy examination is the removal of organs and sutures for histological examination. In burn deaths, histological examinations can demonstrate all degrees and forms of thermal damage to the epidermis [22], such as peeling of the epidermis, intra- and subepidermal cleavage, palisading of basal epithelial cells, and thermal coagulation necrosis from the dermis to subcutaneous tissue and muscle. Connective tissue fibers may also be homogenized and enlarged. In addition, the destruction of nuclear chromatin may be observed, such as the accumulation of adipose tissue in the subcutis, the presence of pseudocysts in the epidermis and dermis, and heat-induced denaturation of intravascular proteins, possibly associated with cellular debris, clots, homogenization, and microthrombi.

To assess lesion viability, a massive intravascular concentration of Sudan III is used, as positive fat emboli may be considered a sign of viability. Remarkably, studies investigating the intravascular concentration of fat emboli after experimental application of postmortem heat showed that this finding did not occur, or that it occurred only in low concentrations [23].

Among the most important findings is the presence of fibrin and leukocytes in the contents of the vesicles, which are histologically localized at the subepidermal level with a partially fluid and partially compact content [24].

Inhalation of hot gas or hot air (heat inhalation trauma) causes extensive damage to the epithelium of the respiratory tract, damage that extends to the second and third orders of the bronchi, and injury to the lung tissue [25,26]. As we have shown in our studies, debrided and coagulative necrosis of the mucous membranes in the nasopharynx, larynx, and trachea occurs in the case of immediate death. Signs of viable lesions in the lungs include severe mucosal oedema with lymphocytic ectasia, capillary, and venous hyperemia with microhemorrhages, and interalveolar, interstitial septal, and perivascular pulmonary oedema. In addition, microscopically, a thin layer of soot is often seen on the pulmonary cords. Fine particles of deeply aspirated soot can be detected microscopically in lung tissue. In live flame exposure, lung damage is evenly distributed throughout the lung tissue, whereas in postmortem heat injury, lung damage is concentrated in the peripheral parts of the organ.

In burn shock, histologic findings must also be specified. Depending on survival, other internal organs also exhibit heat damage, particularly the kidneys, liver, adrenals, brain, and pancreas. Thus, cellular degradation products can lead to acute renal failure with histologic evidence of protein cylinders in the renal tubules.

In addition to “traditional” microscopic examination, immunohistochemistry is also informative in determining the expression of fibronectin, heat shock proteins 27 and 70 (hsp27 and hsp70), and ubiquitin and tryptase expression in the lung tissue of those deceased by flame exposure [27,28]. It is understood that HIF1-α is a transcription factor produced in response to hypoxic conditions which activates gene expression involved in erythropoiesis, angiogenesis, glycolysis, and the modulation of vascular tone. In our research, HIF-1 was tested in a single case of violent asphyxia after strangulation. The intensity of expression is proportional to the duration of the hypoxic stimulus, while HIF1-α rapidly degraded during resuscitation [29]. In the case of postmortem flame exposure, when reoxygenation was not possible, HIF1-α levels may be higher in the lungs, such as its expression in the pulmonary vessels. In this sense, HIF1-α can still be considered a factor of vitality in cases of mechanical asphyxia, even if there was postmortem flame charring [30].

In particular, our study has shown that almost all direct diagnoses require the performance of an autopsy examination and that histological examinations are crucial in more than half of the cases in which they are performed.

When examining a charred body, conducting a toxicological analysis is essential to determine the actual cause of death (e.g., substance abuse with postmortem flaming) and the likelihood of antemortem action of the flames.

In fact, toxicological studies play a critical role in assessing the viability of cadavers that have been covered with flames. In the first place, the determination of the gas concentration in the blood or in the corresponding tissues makes it possible to distinguish and complete the study of respiratory activity at the time of the fire; in this regard, the study of carbon monoxide occupies a major role. Carbon monoxide is an odorless, colorless, and flammable gas that does not irritate the respiratory tract. Its density is approximately equal to that of air. It is produced by the incomplete combustion of various organic substances and is often released by cigarette smoke, fires, and automobile exhaust. In addition, it is formed endogenously in small amounts by heme catabolism and can increase the saturation of hemoglobin with carboxyhemoglobin (COHb) by up to 0.5%. In patients with blood disorders, COHb levels can rise to 3%. Smokers can reach saturation levels as high as 10%. Saturation levels above 50% are considered life-threatening. In addition, CO can bind to the heme group of cytochrome-c oxidase (an enzyme involved in energy production at the mitochondrial level) and to myoglobin, preventing cells from using the remaining oxygen.

After cessation of carbon monoxide exposure, the COHb level decreases by approximately 30–50% per hour. The test is performed with blood from the heart or large blood, but other biological fluids may be used. To claim with sufficient probability that inhalation of the gas has occurred, the COHb level found in the blood must be greater than 10% [31,32,33,34]. One of the compounds produced in a fire is nitric oxide. In particular, the combustion of polymers containing nitrogen (of natural or synthetic origin) produces nitric oxide or hydrocyanic acid (HCN), depending on the oxygen supply. Fatal cyanide poisoning by inhalation can occur after prolonged exposure to cyanide when the HCN concentration is 90 ppm, and after short exposure when the HCN concentration in air reaches 180–270 ppm.

In our study, in all cases of antemortem action flares with death from carbon monoxide intoxication, fibronectin showed strong positive immunohistochemistry. The agreement between these two data allows us to say that, in the absence of toxicological tests for carbon monoxide research, the positivity of fibronectin can still be considered a valid parameter for viability.

These data are particularly important when the autopsy does not show typical elements of viability during exposure to the flame, such as the presence of soot in the bronchi. Similar considerations can be made for HSP70 and CDC26P, although these are less positive on immunohistochemical staining.

Regarding the practice of toxicological tests, where they are not readily available, it is still recommended that they be conducted on subjects from the local area and not on hospitalized subjects. If a subject has been hospitalized, wash-out times of psychoactive substances should be considered. Considering that the average wash-out time for some harmful substances, such as cocaine, is about 48 h, as well as barbiturates or MSD, the likelihood of positive toxicological tests is significantly reduced if the sampling is carried out after 48 h. Similarly, if the subject has been hospitalized for an extended period and undergone blood transfusion or fluid infusion, the wash-out of toxic substances may be even more rapid. Therefore, we can conclude that in the case of a subject who died from burns with hospitalization of more than 48 h, the performance of toxicological examinations, if not available, can be bypassed.

When toxicological analyzes are not readily available, it is recommended that they be performed on subjects from the immediate vicinity rather than in a hospital. If a subject is hospitalized, wash-out times for psychoactive substances should be considered. Considering that the average washout time for substances, such as cocaine, barbiturates, or MSDs is approximately 48 h, the likelihood of positive toxicological tests is significantly lower if sampling is performed after 48 h [35]. Similarly, if the patient has been in the hospital for a long time and has received a blood transfusion or fluid infusion, the washout of toxic substances may be even faster. Therefore, we can conclude that in the case of a subject deceased from burns with a hospital stay of more than 48 h, the performance of toxicological tests, if not available, can be bypassed.

When a body cannot be identified by personal characteristics (tattoos, physical features, etc.), genetic analysis is essential. The nucleic acid most used for genetic analysis is DNA, which is generally amplified by polymerase chain reactions and sequenced with various primers. Even when postmortem degradation of DNA occurs in the soft tissues of the cadaver, some studies have shown that within the third week it is still possible to find DNA of high molecular weight in many tissues (the best results are obtained with the cerebral cortex) [36,37,38,39,40,41,42].

In the cases we selected, the need for genetic analysis was independent of the degree of burn; therefore, it became indispensable only in cases where it was impossible to identify the person from documents, personal items, or tattoos.

In summary, diagnosing the cause of death in a burned corpse is one of the most difficult tasks for forensic pathologists. First, a burn causes extensive damage to the body, especially to the external surface, making autopsy data less reliable and more difficult to interpret. Second, it is usually difficult to determine whether the burn was antemortem or postmortem. Third, a body recovered dead from the fire scene, even if charred, does not automatically mean that they died from the burns [43].

The causes of death in the studied sample can be schematically divided into “natural causes” and “violent causes” [44,45,46]. In our study, among the “natural” causes, the most common is myocardial infarction, i.e., acute myocardial infarction; in addition, one case of sudden death in a patient with epilepsy (SUDEP) was reported. As for the causes of “violent” death, our case history includes numerous cases of death from polytrauma, mainly from traffic and aviation accidents, and, although in a smaller number of cases, from violent mechanical asphyxia and osteo-visceral injuries due to gunshot wounds.

As we have found, the simple external examination is never reliable because it only suggests the cause of death and in most cases superficially suggests lesions due to exposure to flame.

The results obtained have led us to conclude that it is essential to establish and apply a standardized, multidisciplinary protocol that is consistently used in the discovery of bodies resulting from exposure to flame in order to properly address the case from the beginning [47]. The complexity that characterizes the examination of a charred body includes the inescapable need to follow a protocol to avoid the loss of details and important findings that are useful not only for delineating the cause of death but also for identification purposes [48]. In our case history, examination at CT mainly provided information about lesions caused by heat as well as independently (traumatic and by gunshot), and in one case allowed identification by visualization of breast implants [49]. Finally, the immunohistochemical examination was performed mainly in the presence of histological and toxicological results that were not clearly diagnostic (in the presence of equivocal histological images in which no significant soot deposits could be detected, that is accompanied by carboxyhemoglobin levels close to the value described in the literature as a threshold for active gas aspiration) or to provide further evidence of the absence of vitality and, therefore, of postmortem charring [50,51,52,53,54].

In our study, we found that a more complete diagnosis was obtained in all those cases in which all the steps described above were applied, i.e., in which inspection, CT or radiographic examination, autopsy examination, and toxicological, genetic, and histological examinations were systematically performed [55]. This undoubtedly indicates the importance of a systematic, orderly, and recorded investigation of charred or heat-related injury cases to minimize errors or contamination at the scene and to obtain an accurate and rapid diagnosis of the cause of death.

As for the two cases that were not diagnosed, it must be said that their number is too small to produce meaningful statistics. Nevertheless, it can be pointed out that the help of PMCT and toxicology has a high value when an autopsy is performed instead of an external examination. Therefore, as mentioned above, the performance of radiological and toxicological examinations is essential if a direct diagnosis is to be made.

The PMCT and toxicologic studies made it possible to determine with certainty the cause of death in more than half of the cases in which they were used. Since the number of direct diagnoses (DD) and indirect diagnoses (ID) is the same (40 cases in each group), it can be concluded that radiological and toxicological examinations together were used twice as often in DD group cases.

This result suggests that the two tests play a direct role in diagnostic accuracy and prove to be reliable.

Of course, both require a supporting autopsy examination. External examination alone is useful in determining macroscopic viability, but even when combined with toxicologic or radiologic examination, it rarely allows for a definitive diagnosis.

One of the main limitations of our study was that not all tests, such as toxicological or genetic tests, were performed in every case. In fact, to perform the different analyzes, the authorization of the prosecutor is always required.

Similarly, a forensic examination is not always ordered, since the performance of an autopsy is often replaced by the external examination of the corpse alone.

Therefore, it is important to clarify that only completeness in the diagnostic phase makes the results reliable and, above all, standardizable. Hence, there is a need to establish a reliable protocol for all forensic pathologists in cases of charring and to guide the investigation (Figure 14).

The advantages of the proposed protocol are the speed of diagnosis and the ease of application. In addition, we have shown that some tests, such as histological and toxicological examinations, can help determine whether flame exposure occurred before or after death. Finally, tests, such as axial tomography, are essential, especially when violent death is suspected, for example, because of gunshots. Even if laboratories are sometimes not available, it is advisable to perform most examinations, since, as can be seen, they allow satisfactory results to be obtained.

## 5. Conclusions

In summary, based on the results of the study of 82 partially or charred corpses, we have established a protocol based on a series of steps that should be consistently performed to obtain a specific and rapid diagnosis of death.

Our study has made it possible to determine the basic steps in the examination of charred bodies and when they are necessary. These are as follows:Crime scene investigation is always appropriate;Radiologic examination (PMTC or radiography) should be performed in all cases where gunshot wounds, explosions, trauma, charred remains, or the bodies of children are suspected;An autopsy should always be indicated from second-degree injuries;Histology and immunohistochemistry are always required when there is doubt about post-fatal charring;Toxicological analysis is imperative to prove or confirm that the person was alive at the time of the charring, or when the previous steps have led to the suspicion of acute intoxication. Long hospitalizations may result in unusable results;Genetic analysis should be performed when the previous steps fail to identify the body or when violence is suspected.

Specifically, because of the increased specificity of the diagnosis, it is recommended that both radiological and toxicological examination be performed whenever possible.

In the end, we provide a protocol that is accessible and easily searchable by all forensic experts, and which can also be submitted to the judicial authority for approval of certain analyses.

## Figures and Tables

**Figure 1 diagnostics-12-01986-f001:**
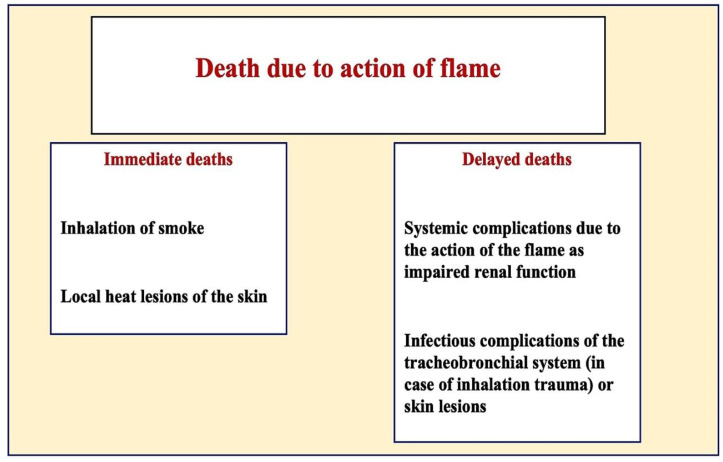
The flowchart illustrates the method for selecting cases of interest from all potentially identified cases.

**Figure 2 diagnostics-12-01986-f002:**
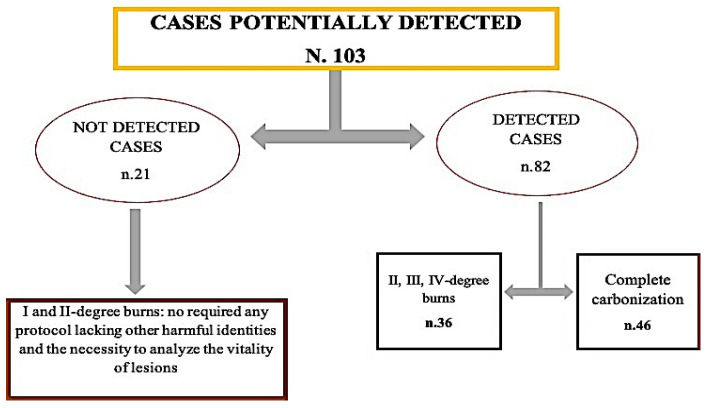
The diagram presents the selection method from a total of 103 cases. Only those belonging to the second group [82 cases] were selected for our study.

**Figure 3 diagnostics-12-01986-f003:**
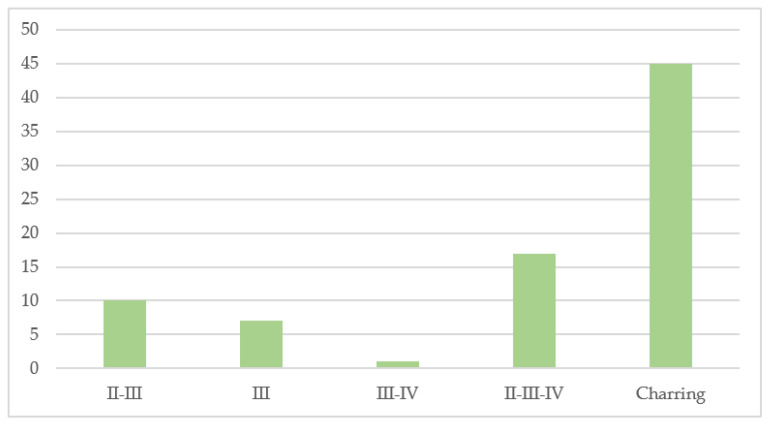
The graph shows the number of subjects who were found charred or with different degrees of burns (II–III; III; III–IV; II–III–IV).

**Figure 4 diagnostics-12-01986-f004:**
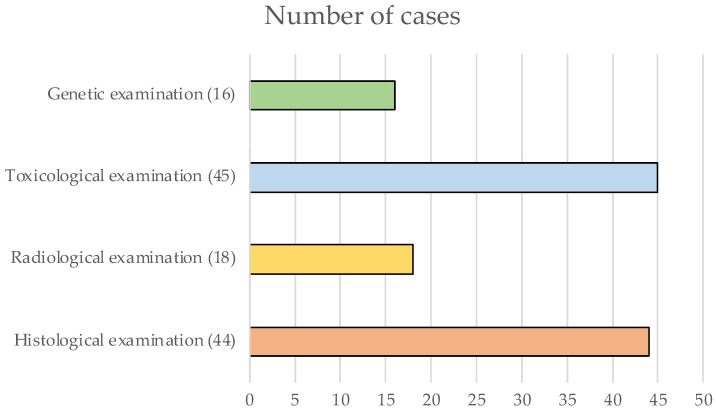
The graph shows the number of analyses conducted by exam type.

**Figure 5 diagnostics-12-01986-f005:**
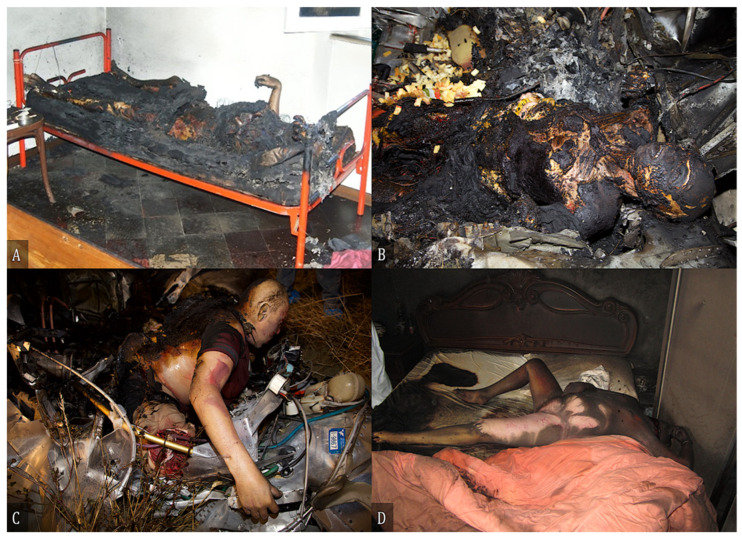
(**A**) Body of a male subject found in an apartment of a building where the fire department had extinguished a fire, in one of the bedrooms. The apartment had been spared the effects of the flames, which had only affected the bed on which the man was lying. (**B**) Body found in the left side of the cockpit of an ultralight aircraft, with the feet stuck in the pedals and the torso stretched towards the rear of the cockpit; the body was almost completely charred and lacked features useful for direct recognition. (**C**) Partially charred body with the lower limbs stuck in the pedals of the aircraft and the torso partially protruding from the cockpit of an ultralight aircraft. (**D**) Body of a female found partially charred in the bedroom of her home.

**Figure 6 diagnostics-12-01986-f006:**
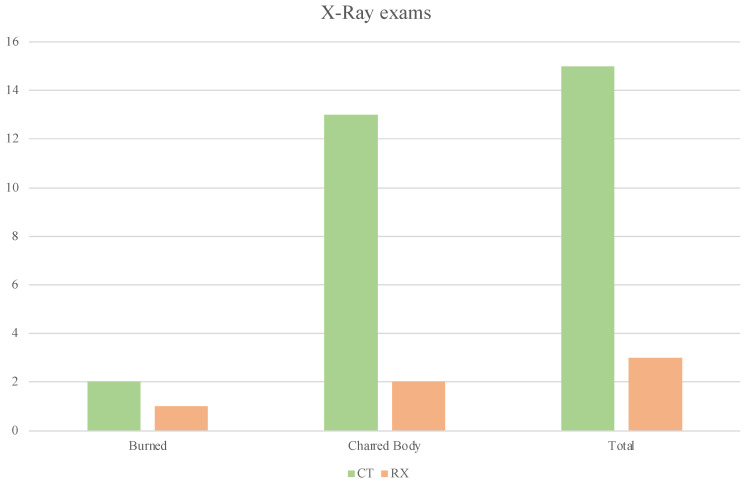
The chart shows the radiological investigations performed on the charred bodies and those with different burn degrees.

**Figure 7 diagnostics-12-01986-f007:**
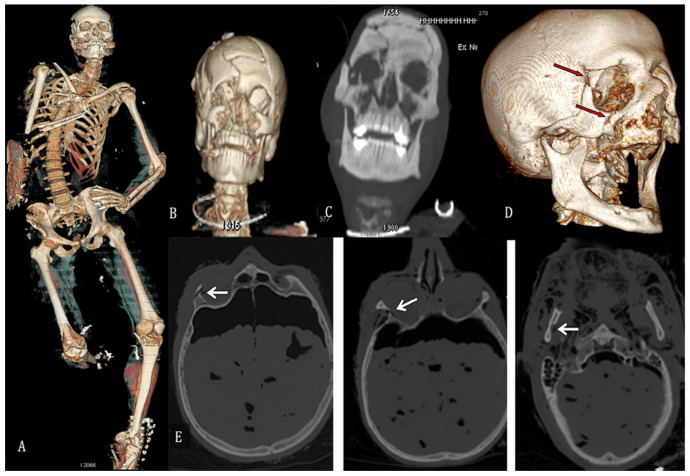
(**A**) Total body CT reconstruction on a partially charred corpse with the following features: heat fracture of the skull, fracture of the mandible, bilateral rib, sternal fractures, L4 and L5 body fractures, left fibula fracture, heat amputation of the right leg. (**B**) A 3D CT whole-body reconstruction in a cadaver with a facial mass fracture. (**C**) A CT whole body with skull and facial massive fractures. (**D**,**E**) Full body scan CT. Fracture of the right orbitofrontal process, fracture of the right zygomatic process, and fracture of the right mandibular body are seen on the skull.

**Figure 8 diagnostics-12-01986-f008:**
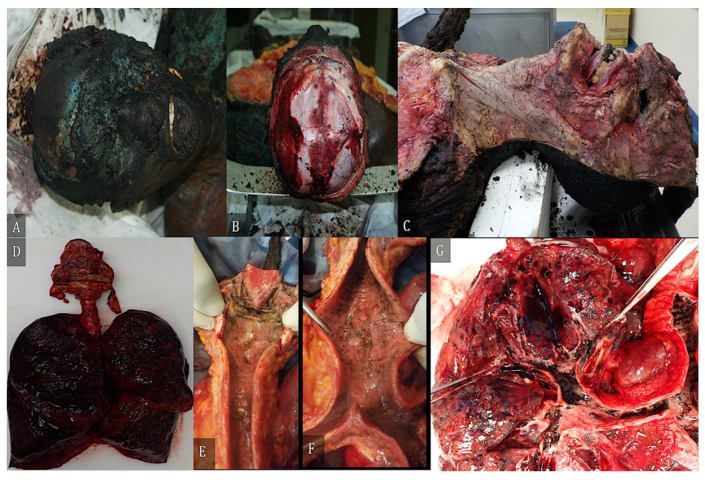
(**A**) Skull of a charred subject, with the tongue, covered with a blackish patina at the tip, protruding from the oral cavity and wedged between the dental arches. (**B**) At the opening of the cranial box, on the inner side of the intact skullcap, we see the presence of a thin extradural hematoma in the left frontoparietal region. (**C**) Layer by layer dissection of the neck that appears non-flamed injured at the level of muscles. (**D**) Example of airway removed “en bloc”, i.e., after isolation of the esophagus and thoracic aorta, careful dissection “en bloc” of the tongue, soft palate, pharynx, hyoid bone, larynx, trachea, bronchi, and lungs was performed. (**E**,**F**) The trachea and bronchi show edematous and hyperemic cherry red mucosa with sooty residue inside; (**G**) Detail of main bronchi showing edematous and hyperemic mucosa and sooty fragments.

**Figure 9 diagnostics-12-01986-f009:**
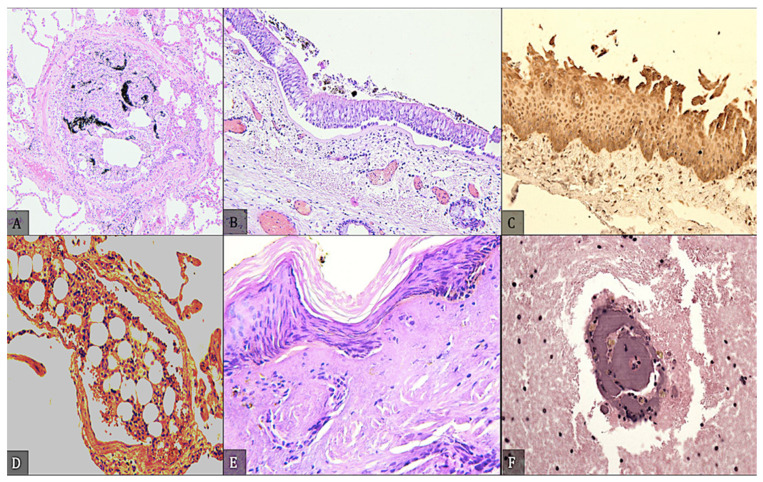
(**A**) Predominantly within the bronchial structures and terminal bronchioles but, also, rather sporadically, in the alveoli, incongruent and blackish-colored particles, sometimes confluent. (**B**) The trachea preparation shows, endo-luminally, the presence of incongruent blackish material, probably sooty and adherent to the epithelium. (**C**) The proximal part of the larynx shows heat epithelial changes well evidenced by the antibody reaction to HSP 27. (**D**) A massive pulmonary adipose embolism demonstrating the viability of the lesion; death occurred because of the burning of the body. (**E**) Peeling of the epidermal layer with the cells arranged as a «palisade» with some sub-epidermal gaps. The underlying connective tissue shows destruction of the nuclear chromatin, as well as phenomena of coagulative necrosis. (**F**) Cerebral vessel with lumen completely occupied by conglutinated blood material due to the action of heat and detachment of the parietal layers.

**Figure 10 diagnostics-12-01986-f010:**
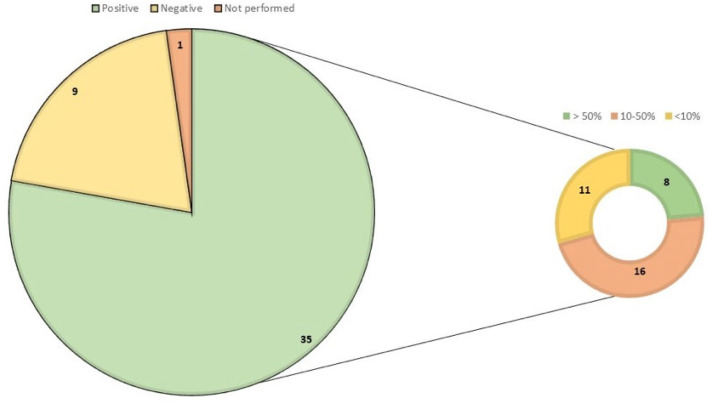
The graph shows the number of cases in which the amount of CoHb in central blood was analyzed. In terms of positive cases for CoHb, the graph on the right shows how many cases were detected in three different percentages, namely below 10%, between 10 and 50%, and above 50%.

**Figure 11 diagnostics-12-01986-f011:**
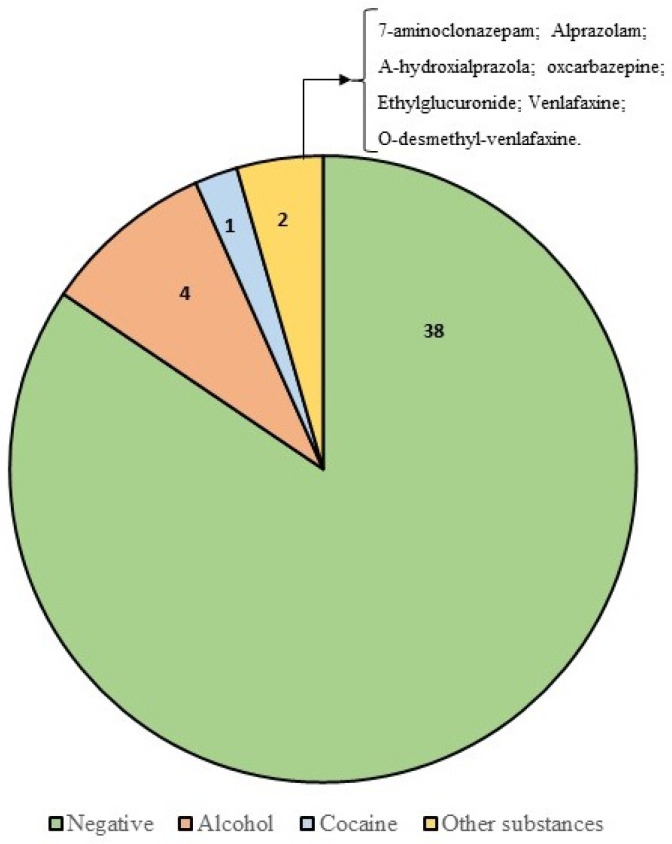
The graph summarizes the results of the toxicological screening. The alcohol concentration in all cases was studied in blood, with 4/45 cases testing positive. Psychoactive substances, such as venlafaxine and benzodiazepines, were found in two cases. Table 5 summarizes the results of the toxicological analysis.

**Figure 12 diagnostics-12-01986-f012:**
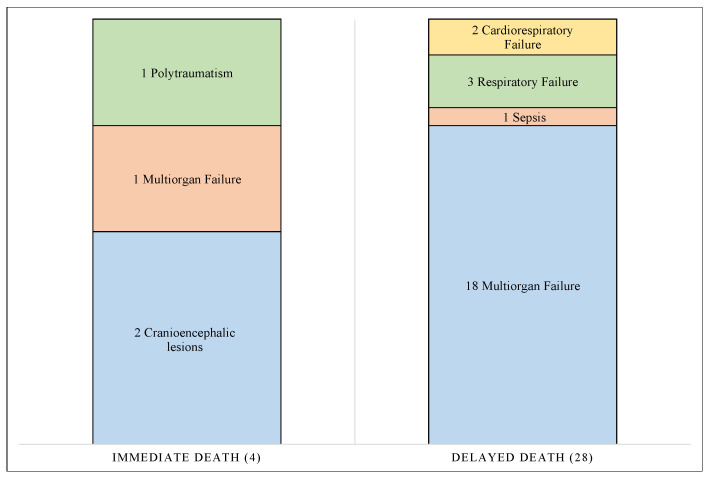
The graph shows all cases of II–III–IV degree burns divided into the following two groups: immediate deaths and delayed deaths. For each group, the cause of death is indicated.

**Figure 13 diagnostics-12-01986-f013:**
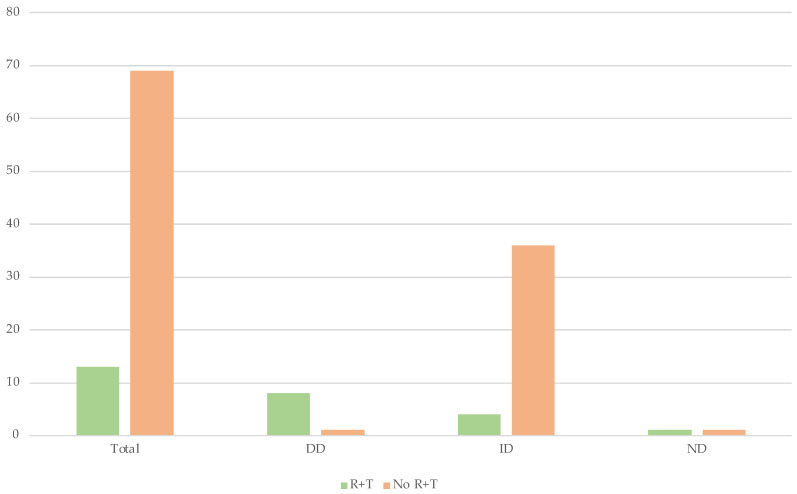
The graph represents all the data described in Table 8.

**Figure 14 diagnostics-12-01986-f014:**
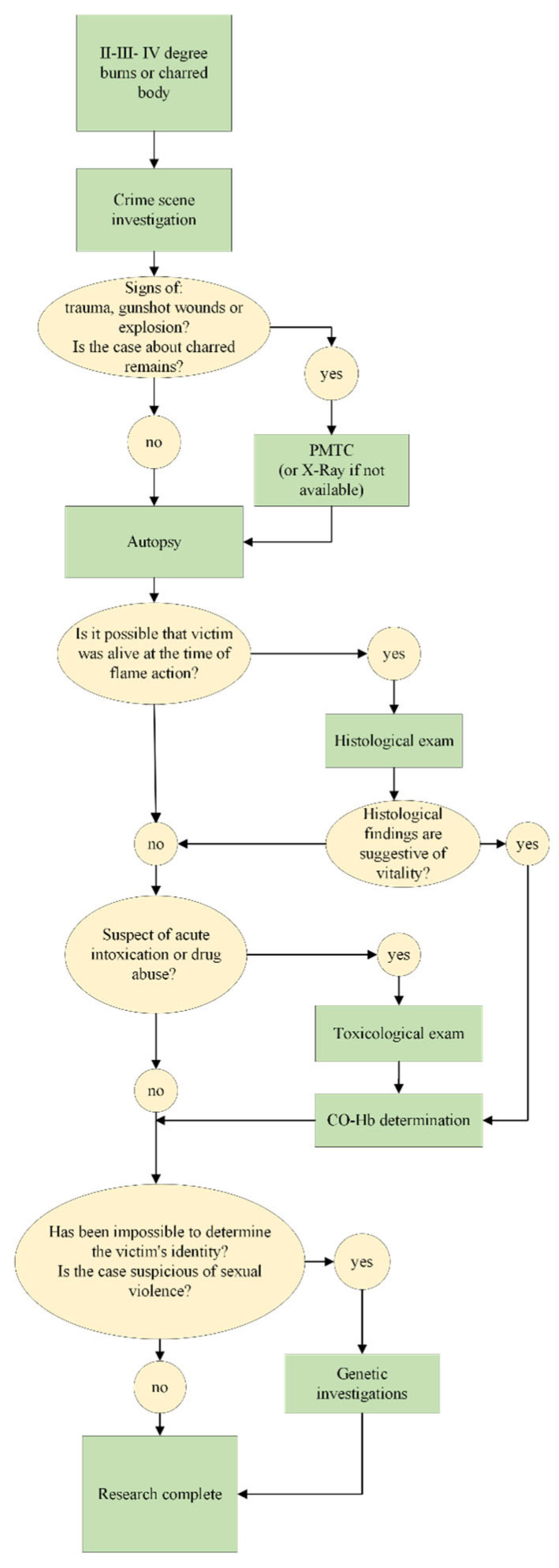
The representation of the proposed protocol to follow during the study of a charred body.

**Table 1 diagnostics-12-01986-t001:** The graphical schematization of the entire case histories analyzed. In each case, each degree of burn is described. It also indicates whether an autopsy or external inspection was performed, and whether other analyses (histological or toxicological) were required. In the column for the type of heat injury, the degree of burn is indicated (II to IV).

Case	Type of Heat Injury	Forensic Inspection	Type of Forensic Exam	Histological Examination	Radiological Examination	Toxicological Examination	Genetic Examination
**Case 1**	Charring	X	A	X	X		
**Case 2**	Charring	X	A	X		X	
**Case 3**	Charring	X	A	X	X	X	
**Case 4**	Charring	X	A	X	X	X	
**Case 5**	Charring		A		X		
**Case 6**	III–IV	X	A	X			
**Case 7**	Charring	X	A	X			
**Case 8**	III		A				
**Case 9**	III	X	EE				
**Case 10**	II–III		A				
**Case 11**	III		EE		X		
**Case 12**	II–III		A				
**Case 13**	Charring		A			X	
**Case 14**	Charring		A	X		X	
**Case 15**	III		A	X			
**Case 16**	Charring		A	X			
**Case 17**	Charring		A	X	X	X	
**Case 18**	Charring	X	A	X	X	X	X
**Case 19**	Charring	X	A	X	X	X	X
**Case 20**	II–III	X	A			X	
**Case 21**	Charring		A		X		
**Case 22**	Charring	X	A		X	X	
**Case 23**	II–III	X	A			X	
**Case 24**	Charring	X	A	X	X	X	
**Case 25**	Charring		A	X		X	
**Case 26**	Charring		A	X		X	
**Case 27**	Charring		A			X	
**Case 28**	Charring	X	A			X	X
**Case 29**	III		A			X	
**Case 30**	III		A			X	
**Case 31**	II–III		A				
**Case 32**	II–III	X	A			X	
**Case 33**	Charring		A				
**Case 34**	Charring		A			X	
**Case 35**	Charring		EE				
**Case 36**	III		A			X	
**Case 37**	Charring		A	X		X	
**Case 38**	Charring		A	X	X	X	
**Case 39**	Charring		A	X		X	
**Case 40**	Charring		A	X		X	
**Case 41**	Charring		A	X			
**Case 42**	Charring		A	X		X	
**Case 43**	Charring		A	X		X	
**Case 44**	Charring	X	A		X		
**Case 45**	Charring		A	X		X	X
**Case 46**	Charring		A	X		X	X
**Case 47**	Charring		A	X		X	X
**Case 48**	Charring		A	X		X	X
**Case 49**	Charring		A	X		X	X
**Case 50**	Charring		A	X		X	X
**Case 51**	Charring		A	X		X	X
**Case 52**	Charring		A	X		X	X
**Case 53**	Charring		A	X		X	X
**Case 54**	Charring		A	X		X	X
**Case 55**	Charring		A	X		X	X
**Case 56**	Charring		EE				
**Case 57**	II, III, IV		A	X		X	
**Case 58**	II, III, IV		A				
**Case 59**	II, III, IV		A	X		X	
**Case 60**	II, III, IV		A	X		X	
**Case 61**	II, III, IV		A	X		X	
**Case 62**	II, III, IV		A				
**Case 63**	II, III, IV		EE				
**Case 64**	II, III, IV		EE				
**Case 65**	II, III, IV		EE				
**Case 66**	II, III, IV		A	X		X	
**Case 67**	II, III, IV		EE				
**Case 68**	II, III, IV		EE				
**Case 69**	II, III, IV		EE				
**Case 70**	II, III, IV		EE				
**Case 71**	II–III		EE				
**Case 72**	II–III		EE				
**Case 73**	II–III		EE				
**Case 74**	II–III		EE				
**Case 75**	II–III		EE				
**Case 76**	Charring		A	X			
**Case 77**	Charring	X	A	X	X	X	X
**Case 78**	II, III, IV	X	A	X	X		
**Case 79**	Charring	X	A	X	X	X	X
**Case 80**	II, III, IV	X	A	X	X		
**Case 81**	II–III	X	A				
**Case 82**	Charring		EE	X	X	X	

**Table 2 diagnostics-12-01986-t002:** The table summarizes the results of the forensic medical examination in 22 cases, along with the location where the body was found. Case 6 and case 7 are two bodies found at the same location, so the same forensic examination was conducted.

Case	Site Inspection
Case 1	Completely charred body found lying on its back in a field with its arms, legs, and neck tied with wire.
Case 2	Carbonized body found in a “wrestling position” in a partially burned reed, lying in a bluff near a railroad line.
Case 3	Carbonized body found sitting in the driver’s seat in a car completely destroyed in the front part at a motorway service area. (A3 motorway)
Case 4	Partially charred body found completely naked and lying crossways on its back in its own bed with legs spread and covered and a cut to the neck.
Case 6	Partially charred body found in the fuselage of an ultralight aircraft crashed in a field. The trunk was protruding from the cockpit and the legs were trapped. Near another lifeless body (Case 7).
Case 7	Partially charred body found in the carcass of an ultralight aircraft that had crashed in a field. He was lying on his back with his legs trapped. Near another lifeless body (Case 6).
Case 9	Almost completely charred body found in a field after a helicopter crash.
Case 18	Almost completely charred body found in a car, sitting in the driver’s seat, arms cuffed behind the back with steel handcuffs. Another charred body on the rear seat. The car had numerous pistol and rifle holes. In the adjacent 7 casings (7.65 caliber) and some cartridges (12 caliber) found.
Case 19	Almost completely charred body inside the car, lying on the floor between the front and rear seats. Arms were bound behind the back with duct tape. The legs were stretched out on the passenger seat and the rest of the body was on the floor of the car. The car had numerous pistol and rifle holes. In the adjacent 7 shell casings (caliber 7.65) and some cartridges (caliber 12) found.
Case 20	Almost completely charred body found lying on its back on the bathroom floor. The apartment showed signs of a recent fire.
Case 22	Partially charred body found lying on its back in a “wrestling position” in an industrial unit with fresh burn marks.
Case 23	Partially charred body found lying on his back in his own bedroom. The apartment had fresh burn marks.
Case 24	Partially charred body found in partially burned brushwood covered ground near a roadway.
Case 28	Partially charred body found lying on its back in a “wrestling position” in its own apartment in a country house.
Case 32	Charred body found lying on its back in a charred mattress in a country house. Near the body a half full glass bottle with wine.
Case 44	Charred male body found in a dwelling. The body was lying on its back in a charred bed with the left wrist raised and tied to the headboard. A coin covered the left eye and another coin lay to the side of the right eye.
Case 77	Body segments belonging to a single subject and signed with the letters ABCD were found behind a military aircraft (C 103 J) in a field near a railway line.
Case 78	Body found in the driver’s seat of a military aircraft (C 103 J) with seat belt fastened in a field near the railway line.
Case 79	The fourth lifeless body was found almost completely submerged in liquid material, identified as fuel, near the fuselage of a military aircraft (C 103 J) in a ditch adjacent to the rail line.
Case 80	Another body was found embedded in the fuselage of the military aircraft (C 103 J).
Case 81	The body was found in a field near the access road to an olive plantation, near a completely burnt car. The body, lying on its back, was naked, with its right arm stretched out between the trunk and the pavement and covered with a green cloth. Remnants of green cloth on the back of the neck. Numerous rocks smeared with reddish liquid (blood).

**Table 3 diagnostics-12-01986-t003:** Results of radiological exams in 82 cases of burned or charred bodies. The presence of fracture, parenchymal lesions, and metallic bodies was investigated.

Case	Type of Exam	Instrumental Findings
Case 1	CT total body	Fracture of the right orbitofrontal process; fracture of right zygomatic process and right mandibular body. Multiple rib fractures. Amputation of phalanx I and II of all fingers in both.
Case 3	CT total body	No fractural lesions on the bony structures. No foreign bodies were detected inside the body.
Case 4	CT total body	Negative for bone fractures and parenchymal lesions. Air bubbles and tissue tears at the level of the soft tissues of the neck. Diffuse opacification of the lung parenchyma as in fluid inhalation.
Case 5	CT total body	No fractural lesions on the bony structures. No foreign bodies were detected inside the body.
Case 11	RX head, left hand, and wrist	Determinate age: over 18 years.
Case 17	CT total body	Mandibular fracture bilaterally reduced and contained with metal synthesis. Disconnection and subversion of the structure of encephalic parenchyma. Calcified density images in the brain parenchyma and presacral soft tissues. Metallic body in the soft tissues of the right ear (earring). Diffuse subcutaneous emphysema. Residual lung parenchyma remains on thoracic scans. Widespread reduction in the calcium density of skeletal segments.
Case 18	CT total body	Metallic body in the left supraclavicular region and the left distal humeral stump.
Case 19	CT total body	Metallic body in the left supraclavicular region, in the endothoracic region, in the entire left lung, and in the right anterosuperior iliac spine.
Case 21	CT total body	No fractural lesions on the bony structures or parenchyma.
Case 22	RX total body	No radiopaque foreign matter within the body.
Case 24	CT total body	Bilateral fracture with horizontal course of both superior horns of the thyroid cartilage. Fracture of the lateral end of the right superior horn of the hyoid bone. Air bubbles in the subcutaneous soft tissues of the anterior part of the neck.
Case 38	CT total body	Acute subdural hematoma with a maximum thickness of 12 mm in the posterior region. No fractural lesions on the bony structures.
Case 44	CT total body	Multifragmentary fracture of the right orbit, the left parietal bone in anterior position, the posterior aspect of the right IV rib, the anterior aspect of the left IV, V rib and the right IV rib, and the left hemisoma and ipsilateral pedicle of L4. Metallic body in the left frontoparietal region, the inner bony cortex of the posterior aspect of the right IV rib, and in the right periscapular region along the medial edge of the body of the scapula (bullet). Large contiguous solution on the right and left hemithorax in the parasternal region at the level of the rib fractures with herniation of lung parenchyma and part of the heart. Bilateral massive hemothorax. Bilateral pneumothorax.
Case 77	CT total body	Dismemberment. Fracture of cervical spine, left clavicle, sternum, and bilateral serial rib fractures. Dislocation of left wrist; plurifragmentation of pelvis and diaphyseal fracture of left femur.
Case 78	CT total body	Fracture in the anterior fossa of the skull base, multiple rib fractures in the left hemithorax, fractures of the vertebral bodies of L1 and L4, the transverse apophysis of L3 and L4, left radius and ulna, right femur, and left tibia and fibula. Subarachnoid hemorrhage. Multiple pulmonary and cardiac contusions.
Case 79	CT total body	Jaw fracture, multiple rib fractures, sternum fractures, left scapula fracture, L4 and L5 vertebral fracture, left fibula fracture. Multiple pulmonary contusions and liver and kidney injuries. Loss of substance in the skull and right leg (heat injuries).
Case 80	CT total body	Skull fractures in the right fronto-parietal-occipito-temporal region, in the facial mass, in the left scapula, in the left humerus, rib fractures on the right, fracture/dislocation of the right tibia, and fibula.
Case 82	RX total body	No radiopaque foreign matter within the body.

**Table 4 diagnostics-12-01986-t004:** The table summarizes the autopsy examinations in 66/82 cases of charred bodies.

Case	Autopsy Results
Case 1	In the neck, a copper wire wound with three turns was found, reaching the thorax and abdomen. Removing the copper, a skin sulcus was documented. The I and II finger were amputated by the fire. Additionally, the ankle was tied by copper wire. In the right front- temporal region, there was a bone fracture. No soot was found in the respiratory system.
Case 2	In the larynx, there was a very slight presence of soot. There was an absence of soot in the trachea and bronchi.
Case 3	Presence of heat amputation (radius and ulna, because of the fire). There was soot in the larynx, trachea, and bronchi.
Case 4	In the anterior region of the neck, there was a continuous solution, in an ecchymotic context, which affects all the muscles of the supra-hyoid region, deepening up to the fibromuscular planes and affecting the vascular structures. No injuries were found in the carotid arteries and jugular veins.
Case 5	There were bone fractures in the upper limbs, lower limbs, ribs, and sternum, with evident hemorrhagic infiltration. No signs of smoke inhalation were found in the trachea. The pericardial sac was torn, leading to the exposure of the heart. There was laceration of the heart, with opening of the right ventricular and atrial cavities. The lower third of the left ventricle presented two lacerations. A further full-thickness laceration was found in the ascending aorta.
Case 6	Occipital fracture and multiple rib fractures with hemorrhagic infiltration were found. There were numerous displaced and exposed fractures of legs. There was a rupture of the pericardium, leading to the exposure of the myocardial wall (because of heat). Laceration of the anterior wall of the right ventricle. Two more tears in both atria. Hemothorax.
Case 11	There was an absence of soot in the respiratory tract. A thickened area in the lower lobe of the right lung was found, along with cardiac hypertrophy and diffuse atherosclerosis.
Case 13	Exposure of the intra-abdominal organs was found. Partial amputation of the upper and lower limbs occurred. There was soot in the bronchial branches.
Case 14	The presence of heat fractures was detected.
Case 15	There were multiple fractures of ribs, and multiple subpleural petechiae bilaterally. Heavy lungs, with areas of contusion, were found mostly on the right. A yellowish secretion was present the level of the right bronchial branches.
Case 16	There was a tear in the left parieto-occipital region. A laceration on the second finger of the left hand was found, as was hemorrhagic infiltration of the internal side of the scalp in the parieto-occipital area. There were multiple rib fractures. There was no presence of sooty material in the tracheal lumen. In the lungs, areas of contusion were found at the level of the bilateral mediastinal face and an interscissural hemorrhagic infiltration was found on the right.
Case 17	Upper airway smoking was found. No residual dust or fumes in the distal middle third of the trachea and bronchus.
Case 18	There was exposed thoraco-abdominal viscera, and fractures in the upper and lower limbs. Complete amputation of the bone and muscle structures of the right thigh occurred. In the clavicle, there was a fragment of the entrance of a bullet. In the larynx and trachea, there was little blackish material. In the left limb there was the entrance of a bullet.
Case 19	There was a fracture between D12 and L1, and a complete absence of bones of the chest. The absence of the abdominal wall and the abdominal and pelvic organs was found. Amputation of the upper and lower limbs occurred bilaterally. At the level of the residue of the right clavicle, the entrance of a bullet was found. At the edge of the right ventricle, there was a continuous full-thickness solution of the ventricular wall. In the lower lobe of the left lung and in the right iliac spine, the entrance of a bullet was found.
Case 20	There was abundant sooty material in the oral cavity and respiratory orifices. The epiglottis and bronchi were covered with abundant brownish sooty material.
Case 21	Exposure of thoraco-abdominal organs was found, with extreme friability of soft tissues. There was no organ damage.
Case 22	There was the exposure of the intestinal packet and loss of joint relations bilaterally at the level of the knee. The presence of hyperemia of the epiglottis and sooty material in the airways were found.
Case 23	There was abundant sooty residues mixed with blood on the tongue. Abundant sooty material was mixed with mucus in the larynx. There was soot and blood in the trachea and bronchi.
Case 24	An ecchymotic area was found at the level of the neck. Hematic infarction at the level of the platysma, sternocleidomastoid, and thyroid-hyoid muscles was identified. There was preternatural motility of the right greater horn of the hyoid bone, and of the upper right and left horns of the thyroid cartilage; there was blood infiltration of neighboring tissues. Hematic material and rare soot were found in the respiratory tract.
Case 25	Laceration was found in the head. There was soot in the larynx, trachea, and in the initial portion of the main bronchi. Hyperemic mucosa in large, medium, and small bronchi with the presence of soot was found.
Case 26	There was soot and food material in the esophagus, and blood and soot in the trachea and inside the bronchi.
Case 27	Loss of substance in the right hemisome and the anterior surface of the lower limbs was found, along with protrusion of the intestinal loops. There was a biosseous fracture in the right wrist. At the epiglottis, there was abundant mucus and soot. There was Diffuse mucosal hyperemia of trachea and bronchi, with soot and cherry red blood in lungs.
Case 28	Exposure of the bony structures of the head and of the thoracic and abdominal viscera was found. There was a fracture of the hyoid bone in the absence of blood infiltration. Soot and blood were found in the airways and bronchi.
Case 29	At the opening of the larynx, trachea and bronchi, soot mixed with blood was found. Lungs were red and expanded. There was edema and soot in the hilum and bronchi.
Case 30	Abundant soot mixed with blood was found in the larynx, trachea, and bronchi. Lungs were red and expanded. Edema and soot were found in the hilum and bronchi.
Case 31	No soot was found in the first respiratory tract. Diffuse hyperemia of the larynx was observed, as well as plate occluding lumen of the circumflex coronary and the right coronary. In the lateral wall, there was a discolored area. In the anterior wall, there was an area with hemorrhagic punctuation. The cardiac parenchyma was affected by diffuse areas of adipose infiltration and myocardiosclerosis.
Case 32	There was soot in the airway (larynx, trachea, bronchi, and upper tract of esophagus).
Case 33	There was soot the in airway (larynx, trachea, and bronchi).
Case 34	The mucosa of larynx, trachea, and bronchi presented oedema, hyperemia, and soot. Increased volume of lungs was found, with a cherry red color. In airways the presence of soot and foam was found, along with hyperemic and cherry red colored mucosa of the bronchi.
Case 36	Abundant sooty material was found in the larynx and trachea; there was soot and blood in the bronchi. Bright red lungs were identified, along with oedema and sooty residues in the hilum and bronchial branches. There was oedema and organs congestion, and a bright red coloring of the blood.
Case 37	In the larynx and trachea abundant mucoid material and red mucosa were found. There was no soot in the bronchial branches.
Case 38	No carbonaceous material was found in the larynx, trachea, and bronchi.
Case 39	In the trachea, there was the presence of blackish mucus and partially digested food material. There was no soot in the bronchial branches.
Case 40	There was exposed intestinal skein, as well as coronarosclerosis and left ventricular hypertrophy.
Case 41	A lacerated bruised lesion was found in the left frontal region. Ecchymosis was found in the jugular vein region. There was multiple bruising in both forearms. The distal phalanx of the third finger of the right hand was sub-amputated. In the larynx and trachea, there was abundant soot mixed with mucus and foam. In the bronchi, the presence of blackish color particles was detected.
Case 42	In the occipital, right posterior parietal and right temporal region, there was a bone breach, with the release of encephalic material completely undone by the heat. Exposure of the intestinal skein occurred. There was an absence of deposits of soot in the respiratory tract.
Case 43	Corpse remains consisted of a trunk and limbs. The right lung was charred. There was no soot in the esophagus and trachea.
Case 44	In the left fronto-temporal region, there was a diamond-shaped continuous solution with clear margins. In the right frontotemporal region, there was a millimeter cutaneous incisura. The wick of a candle protrudes between the lips. In the neck, superficial laceration interesting the soft tissues was found. In the head, there was the presence of two holes, affecting the cranial theca, the first in the right frontal region, and the second in the frontal bone on the left. The two frontal hemispheres are affected by lacerative/hemorrhagic lesions in correspondence with the other cranial lesions. There was a massive bilateral hemothorax. At the hemithorax, two large breaches were found bilaterally. On the mediastinum, circular laceration corresponding to a perforative pericardial lesion was found. Hemopericardium occurred. There were lacerative lesion through the right ventricle of the heart. No foreign material was found in the bronchi. At the loops of the tenuous, three circular lacerations with hemorrhagic infiltration of the tissue were found. There was a circular laceration at the level of the mesentery. There was one laceration in the left colonic flexure and a circular laceration between L4-L5. The first bullet was in the right periscapular soft tissues; the second bullet was at the L4 lamina.
Case 45	There was abundant soot in larynx, trachea, and bronchi.
Case 46	Abundant soot was found in the larynx, trachea, and bronchi. Intracranial hemorrhage occurred because of fire. Abundant soot was found in the trachea and bronchi.
Case 47	Rhymes of heat fractures, and complete loss of the jaw occurred. Exposure of the frontal sinuses occurred. There were Multiple IV fractures. Both lungs were coated by the action of heat, with a charred surface. There was soot in the airway, and a compact parenchyma of the section.
Case 48	Soot and foam were found in the larynx and trachea.
Case 49	Skeletonization of the right tibia and femur occurred, along with disarticulation of the left knee. Blackish material was found in the trachea and in the bronchial branches.
Case 50	Diffuse hemorrhagic infiltration of the prevertebral fascia of the lower part of the cervical spine with “burst” fracture of the soma of C7 occurred.
Case 51	There was no soot in the respiratory tract. There was a fracture of D12 and L1.
Case 52	There was soot in the bronchial branches.
Case 53	Skull fractures occurred because of the fire. Externalization of the intestinal skin, the liver, and the right lung occured. Mutilated upper limbs were observed. Fracture of lower limbs occurred because of fire. Lungs are coated and charred, and there was a “cooked” parenchyma; there was soot in the bronchi.
Case 54	Externalization of the thoraco-abdominal viscera occurred. There were multiple heat fractures in the limbs. Soot was found in the trachea. In the bronchi, rosacea foam and traces of soot were found.
Case 55	No soot was found in the respiratory tract. In the esophagus, larynx, and trachea, a minimal amount of gastric contents was found. Laceration of the left hemidiaphragm occurred.
Case 57	Absence of the distal phalanx of the index finger of the right hand and of the distal portion of the last phalanx of the middle finger was observed. The larynx, esophagus, and trachea were injury-free.
Case 58	The larynx, esophagus, and trachea were injury-free.
Case 59	There was nothing to the esophagus, trachea, and larynx. There was an asence of soot in the bronchi. A congested and edematous lung parenchyma was observed.
Case 60	There was no soot in the bronchi. A congested and edematous lung parenchyma was observed.
Case 61	There was no soot in the bronchi. A congested and edematous lung parenchyma was observed.
Case 66	There was no soot in the bronchi. A congested and edematous lung parenchyma was observed.
Case 76	Soot was found in small quantities in the larynx and trachea. Coerced and charred lungs were observed. There was a blackish punctuation of the parenchyma in the most spared areas.
Case 77	Residues of body segments consisting of the head, neck, thorax, upper limbs, lower limbs, and abdominal-pelvic visceral parts were observed. There was disruption of the skull, with the absence of brain structures and the loss of somatic features. Wide continuous solution to the face and skull was observed, and there was continuous solution in the left mandibular region. A fragmented face occurred. There was a left sterno-costal fracture focus with diastatic fracture of the sternum in the middle third. Multi-fragmentation of the clavicle and all the ribs of the thorax occurred. Multiple lung lacerations were observed. There was continuous endo-myocardial solution in the left ventricle. There was preternatural mobility of the cervical spine in C6–C7; section at the body of D12. The rupture of the bladder, prostate, sigmoid, and tract of colon and small intestine occurred.
Case 78	Multiple bilateral rib fractures with hemorrhagic infiltration of the soft tissues occurred. Laceration of the intercostal muscles and the left parietal pleura was observed. In the heart, there were multiple contusive subepicardial areas in the upper third of the right ventricle and diffuse epicardial hemorrhagic punctuation in the atrial and ventricular area at the base of the right auricle, endocardial laceration, with hemorrhagic infiltration. Paravertebral hemorrhagic infiltration occurred, as did a fracture of L1.
Case 79	In the head, the presence of the burning of the dura with stratification of blood material was found. No carbon residues or blood were found inside the trachea. There were multiple bilateral rib fractures. There was a fracture of the body of the sternum. Bilateral hemothorax occurred, as did “Cooked” lungs.
Case 80	Multiple fractures in the skull were found. Dura mater was lacerated in the fracture. In the brain, blunt hemorrhagic focus in the right hemisphere occurred, as did skull base fractures. There were multiple bilateral rib fractures with blood infiltration of soft tissues.
Case 81	The brain had a “cooked” appearance. There was soot in the trachea. The heart had increased consistency and volume; the common trunk of the left coronary artery had a stenosis of about 50% of its caliber, as did the proximal tract of the anterior descending; the right coronary artery had an eccentric stenosis of 50% of its caliber proximally, as well as marked left ventricular hypertrophy and right ventricular dilatation. In the lungs there was the presence of soot in the bronchi.

**Table 5 diagnostics-12-01986-t005:** Table resumes histological and immunohistochemical findings performed in a total of 60/82 cases. [N] indicates that analyses were performed on neck skin, not in the lung, while [S] means “skin face”.

Case	Histology	Immunohistochemistry
Case 1	In the lungs, alveolar hyperdistention, rupture of alveolar walls, and moderate interalveolar oedema occurred. In the striated muscle tissue of the cervical region, there were areas of dissociation of fibers because of extensive hemorrhagic spread. Fragments of skin and subcutaneous tissues showed rare leukocyte elements and red blood cells.	Fibronectin −; hsp 70 +/−, CD62P −.
Case 2	In the heart, a large-scale connectivity replaces large muscle tracts, with residual myocyte islands embedded in this connectivity-like tissue. Fibroblasts and newly formed capillaries are seen within this loose connective tissue. The presence of neutrophilic granulocytes with both vascular and interstitial margins was observed. Microcirculation characterized by diffuse and mediointimal sclerosis with luminal stenosis that assumes aspects of severe functional criticality was observed.	
Case 3	In the lungs, there was a widespread presence of large optic voids, secondary to fusion of multiple alveoli by the rupture of alveolar septa. A blackish, anthracotic, inert, and powdery material is observed inside some alveoli, which is also present inside foamy macrophages. The bronchial lumen is occupied by soot particles mixed with amorphous eosinophilic material, as well as the decay of the lining epithelium.	
Case 4	In the lungs, focal interstitial oedema, capillary congestion, and hemorrhagic extravasation were observed in the presence of some neutrophilic granulocytes mixed with histiocytic elements in the alveolar spaces.	Fibronectin −; hsp 70 −, CD62P +/−.
Case 6	In the lungs, the presence of areas of endoalveolar and parenchymal hemorrhage of a contusive nature was observed.	
Case 7	In the lungs, areas of endoalveolar and parenchymal hemorrhage consistent with traumatic genesis were observed.	
Case 14	In the lungs, presence of areas of acute and chronic emphysema with septa, characterized by fissures with edges that look like a “tuft of a brush” and a “drumstick” were observed.	
Case 15	In the lungs, the presence of areas of acute emphysema with septa characterized by fissures with margins that look like a “tuft of brush” were observed. The endoalveolar spaces and the septal spaces were occupied by eosinophilic material studded with red blood cells and white line cells.	
Case 16	Elongation of the cell nuclei of trachea occurred. In the lungs, adipose emboli were present. The endoalveolar spaces and the septal spaces were occupied by very abundant red blood cells, macrophages, and polymorphonuclear cells. Hemorrhages in the liver and spleen were present.	Anti HSP27 ++
Case 17	The neck’s skin shows changes due to autolytic phenomena and normal architecture of the papillary component, with greater preservation of the reticular component and fatty infiltration of the dermis itself.	[N] anti-HSP27 and anti-HSP70 ++, anti-HSP90 +/−, anti-tryptase +/−. [S] thigh: HSP27 −, HSP70 −, HSP90 −, tryptase −. HIF-1α +++.
Case 18	In the lungs, endoalveolar oedema, acute stasis, and heat changes were observed.	
Case 19	In the lungs, autolytic destructive phenomena occurred due to the heat. Pulmonary circulation shows the presence of abundant conglutinated blood material within the lumen, and elective staining for fats (Sudan III) shows complete negativity.	
Case 24	In the lungs, multiple fields of empty optic spaces occurred because of the fusion of multiple alveoli due to rupture of septa that were thinned and stretched. Presence of erythrocytes within the alveolar spaces was observed. There we significant hemorrhagic spurs and the presence of neutrophils in platysma and right sternocleidomastoid muscle. Discontinuity of bone tissue with the dislocation of fragments with hemorrhagic spurs are present in greater right horn of hyoid bone and superior horns of thyroid cartilage	
Case 25	In the lungs, parenchyma shows the presence of a modest, inert, and powdery material of a blackish color. This material is rarely present at the level of the lining epithelium of the alveolar cavities and is mainly contained in numerous foamy macrophages in the intralveolar region.	
Case 26	In the lungs, parenchyma shows the presence of a modest inert powdery material with a blackish color.	
Case 37	In the brain, oedema and cerebral stasis were observed. In the lungs, intense pulmonary stasis, and oedema with signs of acute emphysema were observed. High grade myocardial and hepatic stasis, with aspects of red blood cell adhesions, were also observed.	
Case 38	In the brain, oedema, and parenchymatous congestion occurred. Heat hematoma was observed. In the trachea, epithelial flaking and sporadic inflammatory submucosal lympho-monocytic infiltrates were observed. Bronchi show a cellular sloughing of the epithelium and intense leukocyte infiltration in the submucosa, with some eosinophilic granulocytes. At the cardiac level, there were areas of disseminated myocardiosclerosis with localization at the septal level and in the right ventricle, as well as disarray, myocyte hypertrophy and signs of myofiber dissolution.	
Case 39	In the brain, stasis and edema were observed. In the lungs, severe stasis, acute emphysema, evidence of anthracosis, and chronic bronchitis were apparent. In the heart, stasis and evidence of incipient coronary artery disease were found. Steatosis and hepatitis in a picture consistent with alcoholic liver disease were observed. In the skin, heat damage was of a manifestly post-factitious nature.	
Case 40	In the lungs, areas of chronic emphysema and oedema with overt endoalveolar hemorrhagic aspects were observed. In the heart, myocardiosclerosis and significant coronary atherosclerosis occurred, with foci of coagulation and myocytolysis.	
Case 41	In the lungs, incongruous blackish particles sometimes confluent in bronchial structures were observed in the terminal bronchioles and alveoli. In the trachea, the endoluminal presence of incongruous blackish material was found, adhering to the epithelium.	
Case 42	In the lungs, areas of acute emphysema alternating with areas of coarctation of the parenchyma were found; overt endoalveolar hemorrhagic aspects were present, with evidence of adipose-medullary embolization. In the trachea, there was an absence of incongruous material suggestive of inhaled charcoal particles. There was post-fatal heat damage to all organs.	Fibronectin −; hsp 70 +/−, CD62P −.
Case 43	In the lungs, areas of acute emphysema alternating with areas of coarctation of the parenchyma were observed; there were endoalveolar hemorrhagic aspects; there was evidence of adipose-medullary embolization.	
Case 45	In the lungs, congestion and foci of atelectasis and emphysema were observed. In the trachea, congestion and the presence of blackish material on the mucosal surface were observed.	Fibronectin ++; hsp 70 +/−, CD62P ++.
Case 46	In the lungs, congestion, oedema, and intralveolar hemorrhage were observed.	Fibronectin +; hsp 70 ++, CD62P +.
Case 47	In the lungs, pulmonary parenchyma with congestion and oedema was present.	Fibronectin ++; hsp 70 +/−, CD62P ++.
Case 48	In the trachea, there was a tracheal wall with congestion. In the lungs, there was congestion and alveolar oedema.	Fibronectin +; hsp 70 ++, CD62P +.
Case 49	In the lungs, congestion and alveolar oedema were found.	Fibronectin ++; hsp 70 ++, CD62P ++.
Case 50	In the lungs, congestion, oedema, and alveolar hemorrhage were present; there were micronodules of fibrosis in subpleural area. Intraepidermal blisters in skin were observed.	Fibronectin ++; hsp 70 +/−, CD62P ++.
Case 51	In the trachea, congestion occurred. In the lungs, congestion and oedema occurred.	Fibronectin +; hsp 70 +/−, CD62P ++.
Case 52	In the trachea, there was a deposition of blackish material (soot). In the lungs, there was marked congestion and massive oedema. Hyperkeratosis of the skin and presence of blackish material in the stratum corneum was observed.	Fibronectin +; hsp 70 +/−, CD62P +.
Case 53	In the lungs, marked congestion, oedema, and alveolar hemorrhage were observed.	Fibronectin +; hsp 70 +, CD62P ++.
Case 54	In the heart, scarring of myocardial infarction was present. In the lungs, parenchyma with congestion, endoalveolar oedema and microhemorrhages were observed.	Fibronectin +; hsp 70 +, CD62P +.
Case 55	In the lungs, congestion, oedema and intralveolar microhemorrhages were observed.	Fibronectin +; hsp 70 ++, CD62P ++.
Case 57	In the trachea and lungs, congestion and oedema were present. Subepidermal blistering of skin was apparent.	
Case 59	Polyvisceral congestion was observed.	
Case 60	In the lungs congestion, oedema and alveolar microhemorrhages were apparent. Polyvisceral congestion was observed.	
Case 61	Polyvisceral congestion was observed.	
Case 66	Polyvisceral congestion was observed.	
Case 76	In the lungs, there was a massive edema in smoker’s lung. There was evidence of probable alcohol-related liver disease.	
Case 77	In the lungs, there was an alternation of areas dominated by bleeding events with areas of acute emphysema. There was no lack of peribronchial inflammatory infiltrates. Chronic emphysema was observed. In the skin, there was an almost complete loss of stratum corneum with small hemorrhagic foci immediately below; there was abundant incongruous blackish material adherent to the skin surface (lesion with vital signs). In left thigh, there were muscle interruptions of the fibers with modest signs and hemorrhagic spreads. In the right thigh skin, there was no epidermis, a completely burned dermis; hemorrhagic changes were observed in the underlying fat.	
Case 78	In the lungs, abundant hemorrhages with signs of acute and chronic emphysema were observed. In the school child’s brain, there was a small hemorrhagic focus, probably of blunt origin. The left ventricle of the heart contained areas of contraction band necrosis and hemorrhagic microfoci.	
Case 79	In the lungs, there were alternating areas of edema, hemorrhagic edema, acute emphysema, and atelectasis. In the arterioles, there was embolic material of adipose-medullary origin. In some areas, material of obvious food origin was found within the alveoli. Skin fragments strongly altered by heat exposure with areas of vacuolization were observed.	
Case 80	In the lungs, advanced changes, probably due to heat exposure, with homogenization of the parenchyma were observed. Areas of severe edema, even hemorrhagic, and areas of atelectasis were present. In the heart, advanced changes with homogenization of the parenchyma were apparent.	
Case 81	In the lungs, there was discrete fibrous enhancement with discrete anthracotic imprint. Occasional inflammatory infiltrates of chronic significance were observed. There was acute and chronic emphysema, predominantly in the subpleural region. In the left ventricle myocardium, aspects of marked myocytic hypertrophy were present, along with areas of fibrous replacement of various sizes, and myocytes with vacuolar appearance. Subendocardial contraction band necrosis was observed, as was multi-organ congestion.	

**Table 6 diagnostics-12-01986-t006:** Toxicological exams were carried out in 45/82 cases of burned bodies. The COHb and HCN concentration were tested in central blood, while the research of substance of abuse was carried out in blood, urine, hair, and fragments of tissues (brain, kidney, and liver).

Case	COHb, HCN	Substances of Abuse
Case 2	CoHb: 5–10%	Central blood ethyl alcohol: 0.75 g/L, peripheral blood: 0.69 g/dL, content gastric: 0.70 g/dL.
Case 3	CoHb: 60%	Cocaine in the blood: 5 ng/mL, in the urine: 105 ng/mL; benzoylecgonine: in blood 250 ng/mL; in urine 13,800 ng/mL.
Case 4	-	Central blood ethyl alcohol: 3.18 g/L.
Case 13	CoHb: 60–62%	Negative for each substance
Case 14	CoHb: 28–30%	Negative for each substance
Case 17	CoHb: 9.48%	7-aminoclonazepam in the brain, hair, and pubic hair +; Alprazolam in brain, liver, kidney, hair and pubic hair samples +; α-hydroxialprazolam in the brain, liver and kidney +; Oxacarbazepine in the brain, liver and pubic hair +; Ethylglucuronide in hair and pubic hair samples +.
Case 18	Negative	Negative for each substance
Case 19	Negative	Negative for each substance
Case 20	CoHb: 45.3%	Negative for each substance
Case 22	CoHb: 6%	Negative for each substance
Case 23	CoHb: 56%	Negative for each substance
Case 24	CoHb: 3.9%	Negative for each substance
Case 25	CoHb: 19%	Negative for each substance
Case 26	CoHb: 83%	Negative for each substance
Case 27	CoHb: 48%	Negative for each substance
Case 28	CoHb: 22%	Negative for each substance
Case 29	CoHb: 53%	Negative for each substance
Case 30	CoHb: 49%	Negative for each substance
Case 32	CoHb: 60%	Central blood ethyl alcohol: 3.5 g/L
Case 34	CoHb: 56%	Negative for each substance
Case 36	CoHb: 63%	Negative for each substance
Case 37	Negative	Negative for each substance
Case 38	Negative	Venlafaxine, O-desmethyl-venlafaxine: positive.
Case 39	Negative	Central blood ethyl alcohol: 3 g/L
Case 40	Negative	Negative for each substance
Case 42	Negative	Negative for each substance
Case 43	Negative	Negative for each substance
Case 45	CoHb: 12.79%; HCN: 0.78 mg/L	Negative for each substance
Case 46	CoHb: 31.87%; HCN: 1.11 mg/L	Negative for each substance
Case 47	CoHb: 2.96%; HCN: 1.50 mg/L	Negative for each substance
Case 48	CoHb: 11.9%; HCN: 0.91 mg/L	Negative for each substance
Case 49	COHb: 14.3%. HCN: 1.07 mg/L	Negative for each substance
Case 50	COHb: 11.16%; HCN: 1.08 mg/L	Negative for each substance
Case 51	COHb: 6.0%; HCN: 1.09 mg/L.	Negative for each substance
Case 52	COHb: 10.88%; HCN: 0.95 mg/L.	Negative for each substance
Case 53	COHb: 21.42%; HCN: 1.57 mg/L.	Negative for each substance
Case 54	COHb: 1.23%; HCN: 1.44 mg/L	Negative for each substance
Case 55	COHb: 13.9%; HCN: 1.09 mg/L	Negative for each substance
Case 57	COHb: 4.72%; HCN: 1.43 mg/L	Negative for each substance
Case 59	COHb: 11.35% HCN: 1.08 mg/L	Negative for each substance
Case 60	COHb: 12.00%; HCN: 1.23 mg/L	Negative for each substance
Case 61	COHb: 6.13%; HCN: 1.11 mg/L	Negative for each substance
Case 66	COHb: 5.27%; HCN: 1.14 mg/L	Negative for each substance
Case 76	COHb: 1.6%;	Negative for each substance
Case 82	Negative	Negative for each substance

**Table 7 diagnostics-12-01986-t007:** Another graphical schematization of the deduced causes of death in every case and the type of exam carried out. It details cases of largely charred corpses and specifies whether death occurred before (while alive) or after (postmortem) the fire event. Here, A/EE means autopsy or external examination; R means radiological investigations; H means histological examination; T means toxicological examination; G means genetic investigation; AM means antemortem; PM means post mortem; I means immediate; D means delayed.

Case	Cause of Death	Exams Performed	Flame Action	Timing of Death
**Case 1**	Violent mechanical asphyxia consistent with strangulation due to compression of the neck, associated with a large multiple blunt fracture trauma	F-A-H-R	PM	
**Case 2**	Acute myocardial infarction at the level of the lateral portion of the left ventricle, in a patient with severe impairment of the myocardial microcirculation	F-A-H-T	PM	
**Case 3**	Acute CO intoxication with diffuse body charring from flame exposure	F-A-H-R-T	AM	
**Case 4**	Acute carbon monoxide poisoning associated with extensive body charring from flame exposure, and acute methemorrhagic anemia from slaughter, associated with violent mechanical asphyxia from internal submersion by inhalation of blood in the respiratory tract, in a subject in a state of acute alcohol intoxication	F-A-H-R-T	PM	
**Case 5**	Polytraumatism	F-A-H-R-T	PM	
**Case 6**	Polytraumatism	A-R		I
**Case 7**	Polytraumatism	F-A-H	PM	
**Case 8**	Pathophysiological consequences of deep burns affecting 18% of the body surface area	F-A-H		D
**Case 9**	Acute respiratory failure in a patient with burn shock and burns III, involving 90% of body surface area	A		D
**Case 10**	Multi-organ failure in patients with II, III (and IV)-grade burns on almost all body surface area	F-EE		D
**Case 11**	Pathophysiologic sequelae of deep burns affecting 95% of body surface area	A		I
**Case 12**	Multi-organ failure in patients with II, III, and IV-grade burns on almost all body surface area	EE-R		D
**Case 13**	Acute cardio-respiratory failure due to charring on a living object after a burn	A	AM	
**Case 14**	Acute cardio-respiratory failure due to burns (I-, II-, III-degree) and charring of the head, trunk, and four limbs	A-T	AM	
**Case 15**	Septic shock with multiple organ dysfunction syndrome secondary to burns (III degrees) affecting 20–29% of the body surface	A-H-T		D
**Case 16**	Pathophysiological consequences of deep burns affecting in polytraumatism	A-T	PM	
**Case 17**	Violent mechanical asphyxiation	A-H	PM	
**Case 18**	Osteo-visceral injuries from two gunshots to the neck and left upper extremity	A-H	PM	
**Case 19**	Osteo-visceral injuries due to 3 shots to the thorax and pelvic region	A-H-R-T	PM	
**Case 20**	Acute exogenous intoxication due to inhalation of gasses and vapors in individuals with I- and II-degree burns that extended to more than 50% of the body surface area.	F-A-H-R-T-G		
**Case 21**	Damage caused by thermal energy consistent with direct exposure of the body to flammable liquids	F-A-H-R-T-G	AM	
**Case 22**	Damage caused by thermal energy compatible with direct exposure of the body to flammable liquids	F-A-T	AM	
**Case 23**	Violent mechanical asphyxia (strangulation), performed by a copper rope, associated with a large multiple blunt fracture trauma	A-R		
**Case 24**	Acute myocardial infarction at the level of the lateral portion of the left ventricle, in a patient with severe impairment of the myocardial microcirculation	F-A-R-T	PM	
**Case 25**	Acute exogenous carbon monoxide and hydrogen cyanide intoxication associated with II- and III-degree flame burns extending over large portions of the body surface	F-A-T	AM	
**Case 26**	Violent mechanical asphyxia consistent with strangulation due to compression of the neck	F-A-H-R-T	AM	
**Case 27**	Charring due to flame exposure	A-H-T	AM	
**Case 28**	Acute CO intoxication	A-H-T	AM	
**Case 29**	Acute CO and HCN intoxication with diffuse body charring from flame exposure	A-T		
**Case 30**	Charring due to flame exposure	F-A-T-G		
**Case 31**	Acute CO and HCN intoxication with diffuse body charring from flame exposure	A-T		
**Case 32**	Acute CO and HCN intoxication with diffuse body charring from flame exposure	A-T		
**Case 33**	Acute myocardial infarction	A	AM	
**Case 34**	Acute carbon monoxide intoxication	F-A-T	AM	
**Case 35**	Neurogenic shock from thermal energy, consistent with direct exposure of the soma to flame	A	AM	
**Case 36**	Acute CO and HCN intoxication with diffuse body charring from flame exposure	A-T		
**Case 37**	Charring due to flame exposure	A-R-T	PM	
**Case 38**	Acute myocardial infarction	A-H-R-T	PM	
**Case 39**	Sudden death of an epileptic patient (SUDEP)	A-H-T	PM	
**Case 40**	Acute cardiac pump failure developed on a substrate, characterized by a significant degree of myocardiosclerosis and coronary atherosclerosis	A-H-T	PM	
**Case 41**	Charring due to flame exposure	A-H	AM	
**Case 42**	Polytrauma	A-H-T	PM	
**Case 43**	Polytrauma	A-H-T	PM	
**Case 44**	Osteo-visceral injuries due to three gunshots to the head, chest, and abdomen	F-A-R	PM	
**Case 45**	Acute CO and HCN intoxication	A-H-R-T	AM	
**Case 46**	Acute CO and HCN intoxication	A-H-T-G	AM	
**Case 47**	Acute CO and HCN intoxication	A-H-T-G	AM	
**Case 48**	Acute CO and HCN intoxication	A-H-T-G	AM	
**Case 49**	Acute CO and HCN intoxication	A-H-T-G	AM	
**Case 50**	Acute CO and HCN intoxication	A-H-T-G	AM	
**Case 51**	Acute CO and HCN intoxication	A-H-T-G	AM	
**Case 52**	Acute CO and HCN intoxication	A-H-T-G	AM	
**Case 53**	Acute CO and HCN intoxication	A-H-T-G	AM	
**Case 54**	Acute CO and HCN intoxication	A-H-T-G	AM	
**Case 55**	Acute CO and HCN intoxication	A-H-T-G	AM	
**Case 56**	Not identifiable	EE		
**Case 57**	Multi-organ failure in patients with II-, III-, and IV-degree burns over nearly the entire body surface area	A-H-T		D
**Case 58**	Cardiorespiratory failure in patients with II-, III-, and IV-grade burns over nearly the entire body surface area	A		D
**Case 59**	Cardiorespiratory failure in patients with II-, III-, and IV-grade burns over nearly the entire body surface area	A-H-T		D
**Case 60**	Multi-organ failure in patients with II-, III-, and IV-grade burns over nearly the entire body surface area	A-H-T		D
**Case 61**	Multi-organ failure in patients with II-, III-, and IV-grade burns over nearly the entire body surface area.	A-H-T		D
**Case 62**	Acute respiratory failure with shock-lung syndrome and signs of inhalation injury; acute renal failure in patients with II-, III-, and IV-grade burns over nearly the entire body surface area.	A		D
**Case 63**	Multi-organ failure in patients with II-, III-, and IV-grade burns over nearly the entire body surface area.	EE		D
**Case 64**	Multi-organ failure in patients with II-, III-, and IV-grade burns over nearly the entire body surface area	EE		D
**Case 65**	Multi-organ failure in patients with II-, III-, and IV-grade burns over nearly the entire body surface area	EE		D
**Case 66**	Multi-organ failure in patients with II-, III-, and IV-grade burns over nearly the entire body surface area	A-H-T		D
**Case 67**	Multi-organ failure in patients with II-, III-, and IV-grade burns over nearly the entire body surface area	EE		D
**Case 68**	Multi-organ failure in patients with II-, III-, and IV-grade burns over nearly the entire body surface area	EE		D
**Case 69**	Multi-organ failure in patients with II-, III-, and IV-grade burns over nearly the entire body surface area	EE		D
**Case 70**	Multi-organ failure in patients with II-, III-, and IV-grade burns over nearly the entire body surface area	EE		D
**Case 71**	Respiratory failure after escharotomy surgery in patients with extensive II- and III-degree external burns on 80% of the body surface	EE		D
**Case 72**	Multi-organ failure in patients with II-, III-, and IV-grade burns over nearly the entire body surface area	EE		D
**Case 73**	Multi-organ failure in patients with II-, III-, and IV-grade burns over nearly the entire body surface area	EE		D
**Case 74**	Multi-organ failure in patients with II-, III-, and IV-grade burns over nearly the entire body surface area	EE		D
**Case 75**	Multi-organ failure in patients with II-, III-, and IV-grade burns over nearly the entire body surface area	EE		D
**Case 76**	Charring due to flame exposure	A-H	PM	
**Case 77**	Multi-layered traumatic injuries causing body impoverishment	F-A-H-R-T-G	PM	
**Case 78**	Cranioencephalic lesions	F-A-H-R		I
**Case 79**	Mechanical injuries and thermal injuries	F-A-H-R-T-G	PM	
**Case 80**	Cranioencephalic lesions	F-A-H-R		I
**Case 81**	Cardiac failure in persons with deep II- and III-degree burns extending over almost the entire body surface	F-A		
**Case 82**	Not identifiable	EE-R-T		

**Table 8 diagnostics-12-01986-t008:** The table shows the number of cases (and percentage) of subjects undergoing or not undergoing both radiological and toxicological examinations. The data were then matched with diagnostic findings. Here, DD indicates “direct diagnosis”; ID indicates “indirect diagnosis”; ND indicates “undefined”.

Total	R + T			No. R + T		
82	13 (15.9%)			69 (84.1%)		
	DD	ID	ND	DD	ID	ND
	8 (61.5%)	4 (30.77%)	1 (7.9%)	32 (46.4%)	36 (52.2%)	1 (1.4%)

## Data Availability

Not applicable.

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
