# Peer review of "Cause of Death in Charred Bodies: Reflections and Operational Insights Based on a Large Cases Study"

_diagnostics, 2022, doi:10.3390/diagnostics12081986_

Round 1
Reviewer 1 Report
This interesting manuscript demonstrates the experience of analyzing fully or partially charred corpses to offer a proper implementation protocol for determining the cause of death. The chosen topic of work is important and very practical. The most important part of the manuscript is protocol that defines the main steps of a complete diagnostic pathway that pathologists should follow daily in studying charred bodies. The article is very well written and, in my opinion, does not require any corrections. It may be published in the journal.
Some small notice:
- in the verse 230 there is “toral body”, there should be “total body”.
Author Response
Thank you very much for your feedback. We are very happy to hear such positive feedback.
We have corrected the typo in verse 230 ("total" instead of "toral").
Reviewer 2 Report
In this paper, authors are relatively good writing about dead bodies under fire victim with forensic aspect. Except a few textbook section, I think, this paper is a representative guideline to examine the fire death inspection procedure in forensic field.
Some minor revisions are recommended as following comments.
Line 82. Ref. check (Bohnert et al. (1998) [5])
Line 169 Dako protocol needs refernce.
Line 172 Authors selected HIF-1a antibody for immunohistochemistry. As we know, HIF-1a are rapidly expressed in hypoxic condition and disappeared within 1 hour after expression. Would you explained and showed the reference why you choosed this antibody for this study?
Line 187 Genetic investigation: which A-STR kit was used? Power flex fusion or anything?
Line 286 HSP 27 only showed. In your M/M section, so many antibodies are adopted for IHC, but I can't find out any comments in your results section. Needs more detailed interpretation and descriptions in your results section.
Line 286: adipose embolism. how they identified this embolism? I can’t find out frozen section for fat embolism instead of routine H&E processing which is included organic solvent such as xylene.
Table 3 and case 1. Multiple rib fractures -> what is the cause of death in this case?
Line 306 Fig 7. C-D Not good figure quality.
Author Response
Thank you very much for your feedback. We are very happy to receive such valuable feedback.
we have made the correction as below:
In this paper, authors are relatively good writing about dead bodies under fire victim with forensic aspect. Except a few textbook section, I think, this paper is a representative guideline to examine the fire death inspection procedure in forensic field.
Some minor revisions are recommended as following comments.
Line 82. Ref. check (Bohnert et al. (1998) [5])
We have rewritten the reference. We give here the reference found in PubMed:
Bohnert M, Rost T, Pollak S. The degree of destruction of human bodies in relation to the duration of the fire. Forensic Sci Int. 1998;95(1):11-21. doi:10.1016/s0379-0738(98)00076-0.
We selected this article because it contrasts with the findings of Richard et al. on the changes in charred bodies during cremation.
Line 169 Dako protocol needs refernce.
We have added the reference.
Line 172 Authors selected HIF-1a antibody for immunohistochemistry. As we know, HIF-1a are rapidly expressed in hypoxic condition and disappeared within 1 hour after expression. Would you explained and showed the reference why you choosed this antibody for this study?
HIF1-α is a transcription factor produced in response to hypoxic conditions that activates gene expression involved in erythropoiesis, angiogenesis, glycolysis, and modulation of vascular tone. In our research, HIF-1 was tested in a single case of violent asphyxia after strangulation. The intensity of expression is proportional to the duration of the hypoxic stimulus, whereas HIF1-α is rapidly degraded during resuscitation [28]. In the case of postmortem flame exposure, when reoxygenation was not possible, HIF1-α concentrations in the lungs may be higher, such as its expression in pulmonary vessels. In this sense, HIF1-α can also be considered a vital factor in mechanical asphyxia, even if postmortem flame charring has occurred [29].For clarity, we have added this explanation with the appropriate references in the Discussion section (lines 467-475).
Line 187 Genetic investigation: which A-STR kit was used? Power flex fusion or anything?
DNA was extracted using EZ1 Advanced and the EZ1 Tissue Kit (QIAGEN, Germany) and quantified using a spectrophotometer at 260 nm. As a multiple STR amplification kit, our laboratory used PowerPlex® 18D.
We have highlighted it in line 191.
Line 286 HSP 27 only showed. In your M/M section, so many antibodies are adopted for IHC, but I can't find out any comments in your results section. Needs more detailed interpretation and descriptions in your results section.
As suggested, we have added further details on the study of other skin markers and antibodies.
Line 286: adipose embolism. how they identified this embolism? I can’t find out frozen section for fat embolism instead of routine H&E processing which is included organic solvent such as xylene.
Most of the internal organs were so altered that freezing would produce numerous artifacts. For this reason, we decided to examine the findings only by exploiting staining with hematoxylin-eosin. Figure C shows a capillary with numerous optically empty round spaces (white). The alcohol used for staining with hematoxylin-eosin dissolves the lipids, which appear as "white circles" under the microscope. This method is easy to use and very useful, so it was our choice.
Table 3 and case 1. Multiple rib fractures -> what is the cause of death in this case?
In this case ,as reported in table 6, the cause of death is violent mechanical asphyxia associated with asphyxia due to compression of the neck associated with a large blunt fracture trauma. According to our reconstruction, the victim suffered these fractures while struggling with his attacker.
Line 306 Fig 7. C-D Not good figure quality.
Unfortunately, we do not have the original image as it is an archival photograph that was not originally digitized.
Reviewer 3 Report
This study involved a significant amount of work, but the resultant report is not presented in a particularly useful way. The authors should summarize the data, rather than have the reader summarize the data. Most tables should be converted into charts and table 5 should be a graph. The data should be divided for analyses, such as immediate and delayed deaths, deaths where victims died from the fire exposure or died and burned afterwards.
It appears that the primary conclusion (from the abstract) is that "only the application of all the above systematic analyses can provide greater accuracy and reliability in describing the causes of death or solving problems such as identification," but isn't it always true that more data allows greater accuracy and reliability? It would be far more compelling to say that without PMCT only X% were identified compared to Y% without PMCT. There is no statistical evaluation of the data here.
The other main outcome is the so-called "protocol," which seems to be the Figure 8 logical flow chart and the bulleted narration in Section 5 Conclusions. Without the aforementioned statistical data these seem to be mere opinions. As mere opinions, then we wonder with what authority do these authors have to tell everyone else how to do their work? In fact, it is significant that they do not comment on the procedures and protocols of anyone else. I would be more palatable if the authors made recommendations, rather than declaring others should use their protocol.
I disagree with doing radiology only when there is evidence of trauma (including firearms injuries) as I believe that radiology should be done on all burned bodies to rule-out a gunshot wound. In fact, NAME forensic pathology autopsy standards require radiology in all charred remains (G25.4). Nor do I agree with performing toxicology test only when intoxication is suspected or to confirm the person was alive at the time of the fire exposure; I think it should be routine.
Is there anything really new here? We already know that these various procedures may be useful--that is why the Universities in Pisa and Rome did them in the first place. I believe that most offices routinely perform most of these procedures on a routine basis (except the immunohistochemistry, which seems impractical). To the extent that offices do not perform the procedures recommended, they seem to be living with it; perhaps they do not have the resources or perhaps their obtain satisfactory answers without the procedures.
While most of the article uses good grammatical English, there are some issues. "cyanidin should be "cyanide." "flittene" should probably be "bullous formation." The authors use various terms and phrases for fire exposure or extreme heat; instead of asking if the victim was alive at the time of charring, they should ask if the victim was alive at the time of the fire.
Reviewer 4 Report
Dear Author(s),
The paper deals with “Cause of Death in Charred Bodies”. Paper demonstrates the experience of analyzing fully or partially charred corpses and offers a proper implementation protocol for determining the cause of death.
Each case was examined in the systematic of “inspection, radiological examinations, autopsy examination or external examination, histological or immunohistochemical examinations, toxicological examinations, genetic examination”. Therefore, the case evaluation is strong enough to achieve presented results.
There is no need for further comments.
Sincerely,
Author Response
Thank you very much for your opinion. We are glad that our work is complete and useful. We will work to keep it that way.
Round 2
Reviewer 3 Report
The authors have made significant changes that address most of my earlier concerns and I believe the paper has been improved. In particular, the graphs greatly help the presentation. Also, the protocol/conclusions are better supported.
In the caption to Graph 5.1, I would recommend "summarizes" replace "resumes."
Author Response
We thank you again for your important contribution and have corrected the caption to Graph 5.1.
Kind regards